# A Local Particle Filter and Its Gaussian Mixture Extension Implemented with Minor Modifications to the LETKF

Shunji Kotsuki[1,2,3], Takemasa Miyoshi[1,4,5,6,7], Keiichi Kondo[8,1], and Roland Potthast[9,10]

[1]RIKEN Center for Computational Science, Kobe, Japan
[2]Center for Environmental Remote Sensing, Chiba University, Chiba, Japan
[3]PRESTO, Japan Science and Technology Agency, Chiba, Japan
[4]RIKEN Interdisciplinary Theoretical and Mathematical Sciences Program, Kobe, Japan
[5]RIKEN Cluster for Pioneering Research, Kobe, Japan
[6]Japan Agency for Marine-Earth Science and Technology, Yokohama, Japan
[7]Department of Atmospheric and Oceanic Science, University of Maryland, College Park, Maryland, USA
[8]Meteorological Research Institute, Japan Meteorological Agency, Tsukuba, Japan
[9]Deutscher Wetterdienst, Offenbach, Germany
[10]Applied Mathematics, University of Reading, UK

*Correspondence to*: Shunji Kotsuki (shunji.kotsuki@chiba-u.jp) and Takemasa Miyoshi (takemasa.miyoshi@riken.jp)

**Abstract.** A particle filter (PF) is an ensemble data assimilation method that does not assume Gaussian error distributions. Recent studies proposed local PFs (LPFs), which use localization as in the ensemble Kalman filter, to apply the PF for high-dimensional dynamics efficiently. Among others, Penny and Miyoshi developed an LPF in the form of the ensemble transform matrix of the Local Ensemble Transform Kalman Filter (LETKF). The LETKF has been widely accepted for various geophysical systems including numerical weather prediction (NWP) models. Therefore, implementing the LPF consistently
with an existing LETKF code is useful.

This study develops a software platform for the LPF and its Gaussian mixture extension (LPFGM) by making slight modifications to the LETKF code with a simplified global climate model known as Simplified Parameterizations, Primitive Equation Dynamics (SPEEDY). A series of idealized twin experiments were accomplished under the ideal model assumption. With large inflation by the relaxation to prior spread, the LPF showed stable filter performance with dense observations but
became unstable with sparse observations. The LPFGM showed more accurate and stable performance than the LPF with both dense and sparse observations. In addition to the relaxation parameter, regulating the resampling frequency and the amplitude of Gaussian kernels was important for the LPFGM. With a spatially inhomogeneous observing network, the LPFGM was superior to the LETKF in sparsely observed regions where the background ensemble spread and non-Gaussianity are larger. The SPEEDY-based LETKF, LPF, and LPFGM systems are available as open-source software on Github
(https://github.com/skotsuki/speedy-lpf) and can be adapted to various models relatively easily like the LETKF.

**Short Summary.** Data assimilation takes an important part in numerical weather prediction (NWP) to combine forecasted states and observations. While data assimilation methods in NWP usually assume the Gaussian error distribution, some

variables in the atmosphere are known to have non-Gaussian error statistics such as precipitation. This study extended a widely used ensemble data assimilation algorithm for enabling to assimilate more non-Gaussian observations.

## 1 Introduction

Ensemble-based data assimilation (DA) has been broadly applied in geoscience fields such as weather and ocean prediction. The ensemble Kalman filter (EnKF) has been intensely investigated for the past two decades, such as for the perturbed observation method (Evensen, 1994; Burgers et al., 1998; Houtekamer and Mitchel, 1998; van Leeuwen 2020) and ensemble square root filters (e.g., Anderson, 2001; Bishop, 2001; Whitaker and Hamill, 2002; Hunt et al., 2007). The EnKF has advantages in flow-dependent error estimates represented by an ensemble and in its relative ease of implementation to nonlinear dynamical systems such as numerical weather prediction (NWP) models. The degrees of freedom of dynamical models (e.g., $> O(10^8)$ for NWP models) are much larger than the typically affordable ensemble size ($< O(10^3)$). On the other hand, atmospheric and oceanic models show local low dimensionality (Patil et al., 2001; Oczkowski et al. 2005), and practical EnKF implementations use a localization technique that limits the impact of distant observations while also reducing the effective degrees of freedom. Among various kinds of EnKFs, the local ensemble transform Kalman filter (LETKF; Hunt et al., 2007) is widely utilized in operational NWP centers in the same manner with the Deutscher Wetterdienst (DWD) and the Japan Meteorological Agency (JMA). Analysis updates of the LETKF are performed by multiplying the ensemble transform matrix to the prior ensemble perturbation matrix, following the ensemble transform Kalman filter (ETKF, Bishop et al. 2001; Wang et al. 2004).

The particle filter (PF) is another ensemble DA method broadly applicable to nonlinear and non-Gaussian problems (cf. van Leeuwen 2009 and van Leeuwen et al. 2019 for reviews on geoscience applications). The PF potentially solves some issues in the basic assumptions of the EnKF by permitting nonlinear observation operators and non-Gaussian likelihood functions (Penny and Miyoshi, 2016). For example, assimilating precipitation observations with a standard EnKF is difficult partly because of their non-Gaussian errors (Lien et al., 2013, 2016; Kotsuki et al., 2017a). The PF can treat such variables with non-Gaussian errors properly. Several PF methods have been explored for low-dimensional problems in early studies (Gordon et al. 1993; van Leeuwen et al. 2009). However, applying the PF to high-dimensional dynamical systems is generally difficult because the number of particles or the ensemble size must to be increased exponentially with the system size to avoid a weight collapse in which very few particles occupy most of the weights (Snyder et al., 2008, 2015). If a weight collapse occurs, the PF loses the diversity of the particles after resampling. Therefore, we need to assure that the weights are similarly distributed among particles. Previous studies developed the equivalent-weights particle filter (EWPF; van Leeuwen et al. 2010; Ades and van Leeuwen 2013, 2015; Zhu et al. 2016) to extend time until a weight collapse by using the proposal density to drive the particles toward the high-probability region of the posterior. Even PFs with proposal densities, however, are unable to prevent the weights from collapsing.

Alternatively, the local particle filter (LPF) uses localization to avoid a weight collapse by limiting the impact of observations within a local domain. Localization is a well-adopted method for the EnKF to treat sampling errors due to a limited ensemble size. Several LPFs have been proposed to apply the PF efficiently to high-dimensional systems (e.g., Bengston et al., 2003; van Leeuwen, 2009; Poterjoy, 2016; Penny and Miyoshi, 2016; Poterjoy and Anderson, 2016; Farchi and Bocquet, 2018; van Leeuwen et al. 2019). Among them, Penny and Miyoshi (2016) developed an LPF by means of the

ensemble transform matrix of the LETKF. The LETKF has been commonly used for diverse geophysical systems, and consistent implementation with an available LETKF code would be useful. The DWD implemented the LPF in an operational global model based on the operational LETKF code and described stable performance in the operational setting (Potthast et al., 2019). Walter and Potthast (2022; hereafter WP22) extended the LPF to the Gaussian mixture for further improvements.

       The goal of this study is to develop and provide a software platform to accelerate research on LPF and its Gaussian

mixture extension (hereafter, LPFGM). Although many studies on LPF have used simple idealized chaotic models, such as the 40-variable Lorenz model (hereafter, "L96"; Lorenz, 1996; Lorenz and Emanuel, 1998), the present study uses a simplified atmospheric general circulation model defined as the Simplified Parameterizations, Primitive Equation Dynamics (SPEEDY) model (Molteni, 2003). Miyoshi (2005) coupled the LETKF with the SPEEDY model for the first time. This study extends the SPEEDY-LETKF code and implements the LPF and LPFGM. As demonstrated below, the LPF and LPFGM can be

implemented easily with simple modifications to the existing LETKF code.

       The remainder of this paper is organized as follows. Section 2 explains the mathematical formulation of the LETKF, LPF, and LPFGM, followed by the description of specific modifications made to the existing LETKF code. Section 3 presents the experimental settings. Section 4 outlines the results and discussions. Lastly, section 5 provides a summary.

## 2 Methodology

### 2.1 Local ensemble transform Kalman filter

       Hunt et al. (2007) introduced the LETKF as a computationally efficient EnKF by combining the local ensemble Kalman filter (Ott et al., 2004) and ETKF (Bishop et al., 2001). Let $\mathbf{X}_t$ be a matrix composed of $m$ ensemble state vectors. The ensemble mean vector and perturbation matrix of $\mathbf{X}_t$ are given by $\bar{\mathbf{x}}_t$ ($\in \mathbb{R}^n$) and $\mathbf{Z}_t \equiv \left\{ \mathbf{x}_t^{(1)} - \bar{\mathbf{x}}_t, \ldots, \mathbf{x}_t^{(m)} - \bar{\mathbf{x}}_t \right\}$ ($\in \mathbb{R}^{n \times m}$), respectively, where $n$ is the system size. The subscript $t$ indicates the time, and the superscript $(i)$ denotes the $i$th ensemble

member. The analysis equations of the LETKF are specified by:

$$\mathbf{X}_{t,LETKF}^a = \bar{\mathbf{x}}_t^b \cdot \mathbf{1} + \mathbf{Z}_t^b \mathbf{T}_{t,LETKF}, \tag{1}$$

$$\mathbf{T}_{t,LETKF} = \widetilde{\mathbf{P}}_t^a (\mathbf{Y}_t^b)^T \mathbf{R}_t^{-1} \left( \mathbf{y}_t^o - \overline{H_t(\mathbf{X}_t^b)} \right) \cdot \mathbf{1} + \left[ (m-1)\widetilde{\mathbf{P}}_t^a \right]^{1/2}, \tag{2}$$

where $\mathbf{1}$ is a row vector whose all elements are 1 ($\in \mathbb{R}^m$), $\mathbf{T}$ is the ensemble transform matrix ($\in \mathbb{R}^{m \times m}$), $\widetilde{\mathbf{P}}$ is the error covariance matrix in the ensemble space ($\in \mathbb{R}^{m \times m}$), $\mathbf{Y} \equiv \mathbf{HZ}$ is the ensemble perturbation matrix in the observation space ($\in$

$\mathbb{R}^{p \times m}$), $\mathbf{R}$ is the observation error covariance matrix ($\in \mathbb{R}^{p \times p}$), $\mathbf{y}$ is the observation vector ($\in \mathbb{R}^p$), $\mathbf{H}$ is the linear observation operator matrix ($\in \mathbb{R}^{p \times n}$), and $H$ is the observation operator that may be nonlinear. Here, $p$ is the number of observations. The superscripts $o$, $b$, and $a$ denote the observation, background (prior), and analysis (posterior), respectively. The matrix $\mathbf{T}_{LETKF}$ denotes the ensemble transform matrix of the LETKF. The first and second terms of the right-hand side of Eq. (2) correspond to the updates of ensemble mean and perturbation, respectively. The ETKF computes the analysis error covariance matrix in the ensemble space $\widetilde{\mathbf{P}}_t^a$ by:

$$\widetilde{\mathbf{P}}_t^a = \left[ \frac{(m-1)}{\beta} \mathbf{I} + (\mathbf{Y}_t^b)^T \mathbf{R}_t^{-1} \mathbf{Y}_t^b \right]^{-1}, \tag{3}$$

where $\beta$ is a multiplicative inflation factor. Equations (1) and (2) are derived from the Kalman filter equations given by:

$$\bar{\mathbf{x}}_t^a = \bar{\mathbf{x}}_t^b + \mathbf{K}_t \left( \mathbf{y}_t^o - \overline{H_t(\mathbf{X}_t^b)} \right), \tag{4}$$

$$\mathbf{K}_t = \mathbf{P}_t^b \mathbf{H}_t^T (\mathbf{H}_t \mathbf{P}_t^b \mathbf{H}_t^T + \mathbf{R}_t)^{-1} = \mathbf{P}_t^a \mathbf{H}_t^T \mathbf{R}_t^{-1}, \tag{5}$$

$$\mathbf{P}_t^a = (\mathbf{I} - \mathbf{K}_t \mathbf{H}_t)\mathbf{P}_t^b \Leftrightarrow (\mathbf{P}_t^a)^{-1} = (\mathbf{P}_t^b)^{-1} + \mathbf{H}_t^T \mathbf{R}_t^{-1} \mathbf{H}_t, \tag{6}$$

where $\mathbf{K}$ is the Kalman gain ($\in \mathbb{R}^{n \times p}$) and $\mathbf{P}$ is the error covariance matrix in the model space ($\in \mathbb{R}^{n \times n}$). Derivations of Eqs. (5) and (6) are detailed in Appendix A. The EnKF approximates the error covariance matrix by $\mathbf{P} \approx \frac{1}{m-1} \mathbf{Z}(\mathbf{Z})^T$. Hunt et al. (2007) provides more details on deriving Eqs. (1) and (2) from the Kalman filter equations. For nonlinear observation operators, the following approximation is used:

$$\mathbf{Y} \equiv \mathbf{H}\mathbf{Z} \approx H(\mathbf{X}) - \overline{H(\mathbf{X})} \cdot \mathbf{1}. \tag{7}$$

The LETKF provides $\widetilde{\mathbf{P}}_t^a$ and $\left( \widetilde{\mathbf{P}}_t^a \right)^{1/2}$ of Eq. (2) by solving the eigenvalue decomposition of $\left( \widetilde{\mathbf{P}}_t^a \right)^{-1} = \mathbf{U} \boldsymbol{\Lambda} \mathbf{U}^T$, where $\mathbf{U}$ is a square matrix ($\in \mathbb{R}^{m \times m}$) composed of eigenvectors, and $\boldsymbol{\Lambda}$ is a diagonal matrix ($\in \mathbb{R}^{m \times m}$) composed of the corresponding eigenvalues. The eigenvalue decomposition leads to $\widetilde{\mathbf{P}}_t^a = \mathbf{U} \boldsymbol{\Lambda}^{-1} \mathbf{U}^T$ and $\left( \widetilde{\mathbf{P}}_t^a \right)^{1/2} = \mathbf{U} \boldsymbol{\Lambda}^{-1/2} \mathbf{U}^T$.

The localization is practically important for mitigating sampling errors in the ensemble-based error covariance with a limited ensemble size (Houtekamer and Zhang, 2016). With localization, the LETKF computes the transform matrix $\mathbf{T}_{LETKF}$ at every model grid point independently by assimilating a subset of observations within the localization cut-off radius. The LETKF employs localization by inflating the observation error variance so that observations distant from the analysis model grid point have fewer impacts (Hunt et al., 2007; Miyoshi and Yamane, 2007).

For local analysis schemes, a spatially smooth transition of the transform matrix $\mathbf{T}_{LETKF}$ is essential to prevent abrupt changes in the analyses of neighboring grid points. The LETKF realizes a smooth transition of the transform matrix by using the symmetric square root of $\widetilde{\mathbf{P}}_t^a$ (Hunt et al., 2007). The symmetric square root matrix minimizes the mean-square distance

between identity matrix $\mathbf{I}$ and $\left[(m-1)\widetilde{\mathbf{P}}_t^a\right]^{1/2}$; therefore, the analysis ensemble perturbations can be closer to the background ensemble perturbation.

The EnKF generally underestimates the error variance mainly because of model errors, nonlinear dynamics, and limited ensemble size. Therefore, a covariance inflation technique is used to inflate the underestimated error variance (Houtekamer and Mitchell, 1998). Among several kinds of covariance inflations methods, the present study considers multiplicative inflation (Anderson and Anderson, 1999) and relaxation to prior scheme (Zhang et al., 2004) as implemented by Whitaker and Hamill (2012). In multiplicative inflation, the ensemble-based covariance is multiplied by a factor $\beta$ ($\mathbf{P}^b \rightarrow \beta\mathbf{P}^b$). This multiplicative inflation is employed in Eq. (3) in the LETKF. In this study, the LETKF experiments use the approach of Miyoshi (2011) which adaptively estimates a spatially varying inflation factor on the basis of the innovation statistics of Desroziers et al. (2005). In realistic problems, covariance relaxation methods are often used to inflate the posterior perturbation (e.g., Terasaki et al. 2019; Kotsuki et al., 2019b). This study utilizes the relaxation to prior spread (RTPS; Whitaker and Hamill, 2012) given by:

$$\mathbf{Z}_{t(k)}^a \leftarrow \left((1-\alpha) + \alpha\frac{\sigma_{t(k)}^b}{\sigma_{t(k)}^a}\right)\mathbf{Z}_{t(k)}^a, \tag{8}$$

where $\alpha$ is the RTPS parameter, and $\sigma$ is the ensemble spread. The subscript $(k)$ denotes the $k$th model variable or the $k$th component of state vector. Although this study uses only multiplicative inflation for the LETKF experiments, the posterior error perturbation is inflated by RTPS for LPF and LPFGM.

**2.2 Local particle filter with ensemble transform matrix**

Here, we describe the LPF in the form of the ensemble transform matrix of the LETKF (Reich, 2013; Penny and Miyoshi, 2016; Potthast et al. 2019). The PF is a direct Monte Carlo realization of Bayes' theorem given by:

$$\pi(\mathbf{x}_t|\mathbf{y}_{1:t}^o) = \frac{\pi(y_t^o|\mathbf{x}_t)\pi(\mathbf{x}_t|\mathbf{y}_{1:t-1}^o)}{\pi(y_t^o|\mathbf{y}_{1:t-1}^o)}, \tag{9}$$

where $\pi(\mathbf{x}_t|\mathbf{y}_{1:t}^o)$ is the probability of state $\mathbf{x}$ given all observations $\mathbf{y}^o$ up to time $t$. The PF approximates the prior probability density function (PDF), which appears in the numerator of Eq. (9), using an ensemble forecast:

$$\pi_{LPF}(\mathbf{x}_t|\mathbf{y}_{1:t-1}^o) \approx \sum_{i=1}^m w_t^{b(i)}\delta(\mathbf{x}_t - \mathbf{x}_t^{b(i)}), \tag{10}$$

where $\delta$ is the Dirac's delta function and $w^{b(i)}$ is the prior weight of the $i$th particle (i.e., ensemble member) at the previous analysis time. If particles are resampled at the previous analysis time, all particles have the same weight $w^{b(i)} = 1/m$ $(i = 1, \dots, m)$. The sum of the weights $\sum_{i=1}^m w_t^{a,b(i)}$ is always 1. The denominator of the right-hand side of Eq. (9) can be estimated by:

$$\pi(y_t^o|\mathbf{y}_{1:t-1}^o) = \int \pi(y_t^o|\mathbf{x}_t)\pi(\mathbf{x}_t|\mathbf{y}_{1:t-1}^o)d\mathbf{x}_t \approx \sum_{i=1}^m w_t^{b(i)}\pi(y_t^o|\mathbf{x}_t^{b(i)}). \tag{11}$$

This study assumes a Gaussian likelihood function given by:

$$\pi(\mathbf{y}_t^o|\mathbf{x}_t) \propto \exp\left[-\frac{1}{2}\left(\mathbf{y}_t^o - H_t(\mathbf{x}_t)\right)^T \mathbf{R}_t^{-1}\left(\mathbf{y}_t^o - H_t(\mathbf{x}_t)\right)\right]. \tag{12}$$

This assumption means the observation error distribution is assumed to be Gaussian. Then, the posterior can be expressed by:

$$\pi_{LPF}(\mathbf{x}_t|\mathbf{y}_{1:t}^o) \approx \sum_{i=1}^m w_t^{a(i)}\delta\left(\mathbf{x}_t - \mathbf{x}_t^{b(i)}\right), \tag{13}$$

$$w_t^{a(i)} = w_t^{b(i)}\pi_{LPF}\left(\mathbf{y}_{1:t}^o|\mathbf{x}_t^{b(i)}\right)/\left\{\sum_{k=1}^m w_t^{b(k)}\pi_{LPF}\left(\mathbf{y}_{1:t}^o|\mathbf{x}_t^{b(k)}\right)\right\} = w_t^{b(i)}q_t^{(i)}/\left\{\sum_{k=1}^m w_t^{b(k)}q_t^{(k)}\right\}, \tag{14}$$

$$q_t^{(i)} = \exp\left[-\frac{1}{2}\left(\mathbf{y}_t^o - H_t\left(\mathbf{x}_t^{b(i)}\right)\right)^T \mathbf{R}_t^{-1}\left(\mathbf{y}_t^o - H_t\left(\mathbf{x}_t^{b(i)}\right)\right)\right], \tag{15}$$

where $q$ is the likelihood. Equation (14) results in posterior weights that satisfy $\sum_{i=1}^m w_t^{a(i)} = 1$.

To mitigate weight collapse, the *local* PF (LPF) solves the PF equations by assimilating local observations surrounding the analysis grid point. This study uses Penny and Miyoshi (2016)'s approach that computes the analysis at every model grid point independently with the observation error covariance $\mathbf{R}$ being inflated by the inverse of a localization function as in the LETKF (Penny and Miyoshi, 2016).

Sampling importance resampling (SIR) is a technique for applying the PF for high-dimensional dynamics with a limited amount of particles. The resampling process rearranges the particles to effectively represent the densest areas of the posterior PDF. After resampling, each particle has equal weights and the posterior PDF is given by:

$$\pi_{LPF}(\mathbf{x}_t|\mathbf{y}_{1:t}^o) \approx \frac{1}{m}\sum_{i=1}^m \delta\left(\mathbf{x}_t - \mathbf{x}_t^{a(i)}\right). \tag{16}$$

The resampling process can be expressed equally valid to the ensemble transform matrix of the ETKF (Reich, 2013; Penny and Miyoshi, 2016) given by:

$$\mathbf{X}_{t,LPF}^a = \mathbf{X}_t^b\mathbf{T}_{t,LPF} = (\bar{\mathbf{x}}_t^b \cdot \mathbf{1} + \mathbf{Z}_t^b)\mathbf{T}_{t,LPF} = \bar{\mathbf{x}}_t^b \cdot \mathbf{1} + \mathbf{Z}_t^b\mathbf{T}_{t,LPF}. \tag{17}$$

where $\mathbf{T}_{LPF}$ denotes the ensemble transform matrix of the LPF. Here, we applied $(\bar{\mathbf{x}}_t^b \cdot \mathbf{1})\mathbf{T}_{t,LPF} = \bar{\mathbf{x}}_t^b \cdot \mathbf{1}$ based on the following necessary condition of the ensemble transform matrix for the LPF:

$$\sum_{i=1}^m \mathbf{T}_{t,LPF}^{(i,j)} = 1, \qquad j = 1,\ldots,m \tag{18}$$

In addition to Eq. (18), the ideal resampling matrix satisfies the following two conditions:

- $\sum_{j=1}^m \mathbf{T}_{t,LPF}^{(i,j)} = m \cdot w_t^{a(i)}, \qquad for\ i = 1,\ldots,m$
  $\Leftrightarrow \bar{\mathbf{x}}_t^b = \sum_{i=1}^m \mathbf{x}_t^{b(i)} \cdot w_t^{a(i)} \qquad\qquad, \tag{19}$

- spatially smooth transition of $\mathbf{T}_{t,LPF}$ .

where $m$ is the ensemble size, and $\mathbf{T}^{(i,j)}$ indicates the $i$th-row and $j$th-column element of the matrix $\mathbf{T}$.

The resampling matrix significantly affects the filter performance (Farchi and Bocquet, 2018). The present study constructs resampling matrices based on the Monte-Carlo approach. This resampling method uses $m$ random numbers $\mathbf{r}^{(i)}$ ($i = 1, \ldots, m$) drawn from uniform distribution $U([0,1])$ and accumulated weights $w_{t,acc}$:

$$w_{t,acc}^{(0)} = 0, \qquad w_{t,acc}^{(i)} = w_{t,acc}^{(i-1)} + w_t^{a(i)}, \qquad i = 1, \ldots, m. \tag{20}$$

By definition, $w_{t,acc}^{(m)}$ is 1. After sorting $\mathbf{r}$ to be the ascending order $\mathbf{r}_{sorted}$, we generate the resampling matrix using **Algorithm 1**. This procedure is similar to the resampling of Potthast et al. (2019), but it employs an additional treatment so that the resampling matrix is close to the identity matrix (line 10 of **Algorithm 1**; Fig. 1 b and c).

Classical resampling can produce the same particles resulting in the loss of diversity of posterior particles. Therefore, additional treatments are required to generate slightly different posterior particles. Potthast et al. (2019) proposed adding

Gaussian random noise to $\mathbf{T}_{t,LPF}$ to prevent the generation of the same particles. As an alternative, the present study uses the Monte-Carlo approach that repeats **Algorithm 1** many times with different random numbers and takes the average of the generated resampling matrices (Fig. 1). Using different random numbers $\mathbf{r}$, the resampling matrices differ (Fig. 1 b, c) even if the same weight is used (Fig. 1 a). By averaging the resampling matrices generated by using different random numbers, we get matrices that have higher weights in the diagonal components and lower weights in the off-diagonal components (Fig. 1 d,

e). The Monte-Carlo approach makes the resampling matrix closer to the identity matrix, which is beneficial for a smooth transition in space. In addition, this stochastic approach approximates Eq. (19) using the Monte-Carlo approach. The generated transform matrix with 200 samples (Fig. 1 c) is close to that with 10,000 samples (Fig. 1 d) in the case of 40 particles. Therefore, the transform matrices are generated by averaging 200 sampled matrices in subsequent LPF experiments. Hereafter, we call this resampling method "MC resampling" which is used in the following experiments. The number of required samples for

MC resampling is briefly investigated in Appendix B.

The effective particle (or ensemble) size $N_{eff}$ (Kong et al., 1994) is useful to measure the diversity of particles in LPF:

$$N_{t,eff} = 1/\sum_{i=1}^{m} \left(w_t^{a(i)}\right)^2 \in [1, m]. \tag{21}$$

If $N_{eff}$ is sufficiently large ($N_{eff} \approx m$), no resampling is needed. This study considers a tunable parameter $N_0$ as a criterion

for resampling:

$$\mathbf{T}_{t,LPF} \leftarrow \begin{cases} \mathbf{I} & if \quad N_{t,eff} > N_0 \\ \mathbf{T}_{t,LPF} & else \end{cases}. \tag{22}$$

Without resampling ($N_{t,eff} > N_0$), the posterior weight is succeeded for the subsequent forecast as follows:

$$\mathbf{w}_{t+1}^b \leftarrow (1 - \tau) \cdot \mathbf{w}_t^a + \tau/m, \tag{23}$$

where $\tau$ is the tunable forgetting factor $\tau \in [0:1]$. Here, $\tau = 1$ means the subsequent prior weight $\mathbf{w}_{t+1}^b$ has the same weights $1/m$ whereas the posterior weight $\mathbf{w}_t^a$ is completely succeeded to $\mathbf{w}_{t+1}^b$ when $\tau = 0$. This weight succession (Eq. 23) can be interpreted as a temporal localization that reduces the impact of observations temporally distant from the assimilation time. The weight succession in *local* PF is not trivial because the weight (or likelihood of particles) would move with the dynamical flow. Similar discussions can be found for the advection of localization functions (e.g., Ota et al., 2013; Kotsuki et al. 2019a). This study assumes no flow motion of the weights and simply use Eq. (23) at each grid point independently.

Although the LETKF applies inflation to the prior error covariance (i.e., $\mathbf{P}^b \rightarrow \beta \mathbf{P}^b$), this inflation is suboptimal to the LPF partly because the weight collapse occurs more easily if multiplicative inflation is applied to the prior perturbation (cf. Eq. 15). Therefore, the LPF usually applies inflation to the posterior particles (e.g., Penny and Miyoshi 2016; Farchi and Bocquet 2018). The present study uses the RTPS to inflate posterior perturbations for the LPF. We do not use the RTPS for the LETKF experiments since the Miyoshi (2011)'s adaptive multiplicative inflation is known to outperform the manually-tuned RTPS for idealized twin experiments with SPEEDY based on the authors' preliminary experiments and personal communication (Ota et al. 2021).

Preliminary experiments showed that the RTPS outperformed the relaxation to prior perturbation (RTPP; Zhang et al., 2004) for the LPF. Having some noise in the transform matrix is important, so that the LPF can maintain the diversity of posterior particles. The RTPP makes the transform matrix closer to the identity matrix, resulting in less diverged posterior particles. Therefore, the RTPS would be a more suitable relaxation method than the RTPP for LPF.

## 2.3 LPF with Gaussian mixture extension

Assimilating too many independent observations in the local area is not desirable for the LPF to avoid weight collapse (van Leuuwen et al., 2019). To solve this problem, hybrid algorithms of EnKF and PF have been explored to efficiently assimilate massive observation data. The Gaussian mixture extension of the LPF is one such hybrid algorithms (Hoteit et al. 2008; Stordal et al., 2011; WP22). In the Gaussian mixture extension, the prior PDF is approximated by a combination of Gaussian distributions centered at the values of the particles, given by:

$$\pi_{LPFGM}(\mathbf{x}_t|\mathbf{y}_{1:t-1}^o) \approx \sum_{i=1}^m w_t^{b(i)} N(\mathbf{x}_t^{b(i)}, \widehat{\mathbf{P}}_t^b), \tag{24}$$

where $N(\mathbf{x}_t^{b(i)}, \widehat{\mathbf{P}}_t^b)$ is the Gaussian kernel with mean $\mathbf{x}_t^{b(i)}$ and covariance $\widehat{\mathbf{P}}_t^b$. Here, hat indicates matrices for the Gaussian kernels (e.g., prior error covariance $\widehat{\mathbf{P}}^b$, observation error covariance $\widehat{\mathbf{R}}$ and Kalman gain $\widehat{\mathbf{K}}$). The covariance of the Gaussian kernel uses the sampled covariance matrix $\mathbf{P}_t^b$ such that:

$$\widehat{\mathbf{P}}_t^b = \gamma \mathbf{P}_t^b \approx \frac{\gamma}{m-1} \mathbf{Z}_t^b (\mathbf{Z}_t^b)^T, \tag{25}$$

where $\gamma$ (>0) is a tunable parameter that regulates the amplitude and width of the Gaussian kernel (i.e., uncertainty of particles). For example, larger $\gamma$ reduces amplitude and increases width of the Gaussian kernel. Because the kernel is supposed to have a

Gaussian distribution, increasing the Gaussian kernel's width results in a decrease in amplitude. The LPFGM results in the same analysis of the LPF when $\gamma$ is 0 ($\gamma \rightarrow 0$).

The Gaussian mixture performs a two-step update to obtain the posterior particles. The first update moves the center of the Gaussian kernel with observations. Since kernels are Gaussian, we can use the Kalman filter for the first update:

$$\mathbf{x}_t^{a(i)} = \mathbf{x}_t^{b(i)} + \widehat{\mathbf{K}}_t \left( \mathbf{y}_t^o - H(\mathbf{x}_t^{b(i)}) \right), \qquad (i = 1, \ldots, m)$$

$$\Leftrightarrow \mathbf{X}_{t,GM}^a = \mathbf{X}_t^b + \widehat{\mathbf{K}}_t \left( \mathbf{y}_t^o \cdot \mathbf{1} - H_t(\mathbf{X}_t^b) \right), \tag{26}$$

where $\widehat{\mathbf{K}} = \widehat{\mathbf{P}}^a \mathbf{H}^T \mathbf{R}^{-1}$ as in Eq. (5). In the LETKF, the Kalman gain $\mathbf{K}$ is computed by $\mathbf{K} = \mathbf{P}^a \mathbf{H}^T \mathbf{R}^{-1} \approx \mathbf{Z}^b \widehat{\mathbf{P}}^a (\mathbf{Z}^b)^{\mathsf{T}} \mathbf{H}^T \mathbf{R}^{-1}$. Since the Gaussian kernel uses the ensemble-based error covariance (Eq. 25), we can apply the exact algorithms of the LETKF to compute $\widehat{\mathbf{P}}^a$ by replacing $\beta$ of Eq. (3) with $\gamma$ (i.e., $\widehat{\mathbf{P}}^a = \left[ \frac{(m-1)}{\gamma} \mathbf{I} + (\mathbf{Y}_t^b)^T \mathbf{R}_t^{-1} \mathbf{Y}_t^b \right]^{-1}$). Here the same gain matrix $\widehat{\mathbf{K}}$ is applied to update each particle independently while the gain matrix $\widehat{\mathbf{K}}$ is based on the forecast error covariance estimated from the entire ensemble. Consequently, Eq. (26) is equivalent to:

$$\mathbf{X}_{t,GM}^a = \bar{\mathbf{x}}_t^b \cdot \mathbf{1} + \mathbf{Z}_t^b \mathbf{T}_{t,GM}, \tag{27}$$

where

$$\mathbf{S}_t = \widehat{\mathbf{P}}_t^a (\mathbf{Y}_t^b)^T \mathbf{R}_t^{-1} \left( \mathbf{y}_t^o \cdot \mathbf{1} - H_t(\mathbf{X}_t^b) \right), \text{ and} \tag{28}$$

$$\mathbf{T}_{t,GM} = \mathbf{S}_t + \mathbf{I}. \tag{29}$$

Here, $\mathbf{T}_{t,GM}$ denotes the ensemble transform matrix of the Gaussian mixture.

The second update resamples the particles based on the likelihood of the posterior kernel given by:

$$q_t^{(i)} = \exp\left[ -\frac{1}{2} \left( \mathbf{y}_t^o - H_t(\mathbf{x}_t^{b(i)}) \right)^T \widehat{\mathbf{R}}_t^{-1} \left( \mathbf{y}_t^o - H_t(\mathbf{x}_t^{b(i)}) \right) \right], \tag{30}$$

where

$$\widehat{\mathbf{R}}_t = \mathbf{R}_t + \mathbf{H}_t \widehat{\mathbf{P}}_t^b \mathbf{H}_t^T. \tag{31}$$

Hoteit et al. (2008) and Stordal et al. (2011) used Equation (30) for computing the likelihood of posterior kernels. Alternatively, WP22 suggested that using Eq. (15) instead of Eq. (30) is a reasonable approximation in the case of a smaller variance of $\widehat{\mathbf{P}}_t^b$ compared to the observation departure $\mathbf{y}_t^o - H_t(\mathbf{x}_t^{b(i)})$. This study follows the WP22's approximation because computing inverse of Eq. (31) is computationally much more expensive than computing inverse of diagonal $\mathbf{R}$. Using WP22's approximation, the solution of the LPFGM is given by:

$$\mathbf{X}_{t,LPFGM}^a = \mathbf{X}_{t,GM}^a \mathbf{T}_{t,LPF} = \left( \bar{\mathbf{x}}_t^b \cdot \mathbf{1} + \mathbf{Z}_t^b \mathbf{T}_{t,GM} \right) \mathbf{T}_{t,LPF} = \bar{\mathbf{x}}_t^b \cdot \mathbf{1} + \mathbf{Z}_t^b \mathbf{T}_{t,GM} \mathbf{T}_{t,LPF}. \tag{32}$$

Here, we used $(\bar{\mathbf{x}}_t^b \cdot \mathbf{1})\mathbf{T}_{t,LPF} = \bar{\mathbf{x}}_t^b \cdot \mathbf{1}$ (cf. Eq. 18). Consequently, the transform matrix of the LPFGM ($\mathbf{T}_{LPFGM}$) is given by:

$$\mathbf{T}_{t,LPFGM} = \mathbf{T}_{t,GM}\mathbf{T}_{t,LPF}. \tag{33}$$

Namely, the LPFGM can be described as the ensemble transform matrix form. Representing the LPFGM with only one transform matrix $\mathbf{T}_{LPFGM}$ is practically beneficial if one aimed to reduce computational costs by the weight interpolation as for the LETKF in which the transform matrices at higher-resolution model grid points are interpolated by transform matrices at

265 coarser model grid points (Yang et al. 2009; Kotsuki et al. 2020). The weight interpolation is also useful for ensuring spatially smooth transition of the transform matrix for LPFs (Potthast et al. 2019).

The two-step update of the LPFGM may appear to use the same observations twice, but this is not true. To understand the principles, here we consider a simple scalar example with $H$=1.0 with illustrations of Fig. 2. Let $\pi(y_t^o)$ be an observation PDF (Fig. 2 top row), and the prior and posterior PDFs of the LPF are given by $\pi_{LPF}(x_t) \approx \frac{1}{m}\sum_{i=1}^{m} \delta(x_t - x_t^{b(i)})$ and

270 $\pi_{LPF}(x_t|y_t^o)$, respectively (Fig. 2a, 2nd and 3rd rows). The LPF employs resampling by approximating the posterior PDF as *a combination of prior particles* such that $\pi_{LPF}(x_t|y_t^o) \approx \sum_{i=1}^{m} w_t^{a(i)}\delta(x_t - x_t^{b(i)})$ (Fig. 2a, bottom row). Next, we focus on the 5th particle of the LPFGM (Fig. 2b). Prior and posterior PDFs of the 5th particle are given by $\pi(x_t^{(5)}) = N(x_t^{b(5)}, \hat{P}_t^b)$ and $\pi(x_t^{(5)}|y_t^o) = N(x_t^{b(5)}, \hat{P}_t^b) \cdot N(y_t^o, R_t)$, respectively (Fig. 2b, 2nd and 3rd rows). Since the Gaussian kernels are assumed for the prior particles, the center of the posterior kernel moves such that $\pi(x_t^{(5)}|y_t^o) \propto N(x_t^{a(5)}, \hat{P}_t^a)$, where $x_t^{a(5)}$ and $\hat{P}_t^a$ can be

computed by the Kalman filter (from the blue circle to the red circle in Fig. 2b). Since the LPFGM moves all particles, the posterior PDF of the LPFGM is given by $\pi_{LPFGM}(x_t|y_t^o) \approx \frac{1}{m}\sum_{i=1}^{m} \pi(x_t^{(i)}|y_t^o)$ (Fig. 2c, 3rd row). These movements correspond to the first update of the LPFGM. In contrast to the LPF, the posterior PDF of the LPFGM is approximated by *a combination of the posterior Gaussian kernels* (red circles of Fig. 2c, 3rd row). The LPFGM employs the resampling based on the moved particles such that $\pi_{LPFGM}(x_t|y_t^o) \approx \sum_{i=1}^{m} w_t^{a(i)}N(x_t^{a(i)}, \hat{P}_t^a)$ (Fig. 2c, bottom row). This resampling corresponds

to the second update of the LPFGM. As seen in this example, the two-step update of the LPFGM does not use the same observations twice.

## 2.4 Implementation and computational complexity

We implemented the LPFGM based on the available LETKF code from Miyoshi (2005) and follow-on studies (Miyoshi and Yamane 2007, Kondo and Miyoshi 2019; Kotsuki et al. 2020; https://github.com/takemasa-miyoshi/letkf).

Figure 3 compares the workflows of the LETKF, LPF, and LPFGM. All DA methods execute the same first four steps (steps A–D). After step D, the LETKF involves four additional steps (steps E–H) to compute $\mathbf{T}_{LETKF}$. The LPF computes the weights of particles (step J), followed by the generation of the transform matrix $\mathbf{T}_{LPF}$ (step K). The LPFGM first executes steps E–G, as in the LETKF, and then executes step I to compute $\mathbf{T}_{GM}$, followed by the LPF algorithms to compute $\mathbf{T}_{LPF}$ (steps J and K).

Finally, the LPFGM multiplies $\mathbf{T}_{GM}$ and $\mathbf{T}_{LPF}$ to compute $\mathbf{T}_{LPFGM}$ (step L). At the end of the process, the transform matrix is applied to the prior perturbation matrix $\mathbf{Z}^b$ to produce the analysis ensemble (step M), followed by the RTPS (step N).

If the LETKF code is available, the LPF in the transform matrix form can be developed by coding two steps J and K. The LPFGM can also be developed easily by coding two more steps I and L if the LETKF and LPF codes are available.

Here we compare the computational complexities of the LETKF, LPF and LPFGM algorithms (Table 1). The total cost ($C_T$) of a DA cycle is identical to the overhead of the assimilation system ($C_H$), plus $n$ times the average local analysis cost ($C_L$), and $m$ times the cost of one-member model forecast ($C_M$):

$$C_T(n, p, m) = C_H(n, p, m) + n \cdot C_L(p_L, m) + m \cdot C_M, \tag{34}$$

where $p_L$ is the number of local observations within the localization cut-off radius. We assume that the overhead and model costs are equivalent among the three methods. In addition, the total computational cost of DA usually depends on the local analysis cost ($C_L$):

$$C_L^{LETKF} = O(m^3 + m^2 p_L), \tag{35}$$

$$C_L^{LPF} = O(m^2 N_{MC}), \tag{36}$$

$$C_L^{LPFGM} = O(m^3 + m^2 p_L + m^2 N_{MC}), \tag{37}$$

where $N_{MC}$ is the number of times the resampling matrices are generated by **Algorithm 1**. The number of local observations $p_L$ is usually much greater than the ensemble size $m$ and $N_{MC}$. In this case, $O(m^2 p_L)$ is dominant for the LETKF, and LPFGM.

The computational cost of LPF here is more expensive than that with a simpler resampling algorithm such as the stochastic uniform resampling ($O(m)$; Penny and Miyoshi 2016, Farchi and Bocquet, 2018) due to the relatively complex **Algorithm 1** ($O(m^2 N_{MC})$). We could not use such simpler approaches in this study since they do not yield ideal resampling matrices that satisfy the two conditions (Eq. 19 and spatially smooth transition of $\mathbf{T}_{t,LPF}$). The computational cost $O(m^2 N_{MC})$ of LPF is still much smaller than the LETKF if $N_{MC} \ll p_L$.

# 3 Experimental settings

## 3.1 SPEEDY model

This study used the intermediate global atmospheric general circulation model SPEEDY (Molteni, 2003) to compare the LETKF, LPF and LPFGM. The SPEEDY model is a computationally-inexpensive hydrostatic model with fundamental physical parameterization schemes such as surface flux, radiation, convection, cloud and condensation. The SPEEDY model has $96 \times 48$ grid points in the horizontal plane (T30 ~ 3.75° × 3.75°) and sigma-coordinate seven vertical layers. The SPEEDY consists of five prognostic variables: temperature (T), specific humidity (Q), zonal wind (U), and meridional wind (V) at seven layers, as well as surface pressure (Ps). The SPEEDY model coupled with the LETKF (SPEEDY−LETKF) has been widely

used in DA studies (Miyoshi 2005 and many follow-on studies, e.g., Miyoshi 2011; Kondo et al., 2013; Kondo and Miyoshi, 2016, 2019; Kotsuki and Bishop 2022). We implemented the LPF and LPFGM based on the existing SPEEDY−LETKF code following the procedures described in section 2.4.

## 3.2 Experimental design

The experimental settings follow the previous SPEEDY−LETKF experiments (Miyoshi 2005 and follow-on studies, e.g., Kotsuki et al., 2020). A series of idealized and identical twin experiments (also known as observing system simulation experiments) were conducted without model errors. We first performed a spin-up run for one year initialized by the standard atmosphere during rest, followed by the nature run started at 0000 UTC on January 1 of the second year. We assumed diagonal observation error covariance $\mathbf{R}$ (i.e., uncorrelated observation error). Gaussian noise was added to the nature run to produce observation data at 6 h intervals. The standard deviations of the Gaussian noise are 1.0 K for T, 1.0 m s$^{-1}$ for U and V, 0.1 g kg$^{-1}$ for Q, and 1 hPa for surface pressure. This study considered two observing networks (Figs. 4 a and b): the regularly distributed network (hereafter REG2) and the radiosonde-like inhomogeneous network (hereafter RAOB). We observe T, U and V at all seven layers, whereas Q was observed at the 1st−4th layers. The ensemble size is 40 and their initial conditions were taken from an independent single deterministic SPEEDY forecast with sufficient spin-up simulation.

Table 2 summarizes the settings of the LETKF, LPF, and LPFGM experiments. The observation error variance was inflated for the localization, by using the Gaussian-based function given by:

$$l = \begin{cases} \exp\left[-\frac{1}{2}\{(d_h/\rho_h)^2 + (d_v/\rho_v)^2\}\right] & if \ d_h < 2\sqrt{10/3}\rho_h \ and \ d_v < 2\sqrt{10/3}\rho_v \\ 0 & else \end{cases}, \tag{38}$$

where $l$ is the localization function, and its inverse $l^{-1}$ is multiplied and used to inflate $\mathbf{R}$. $d_h$ and $d_v$ are horizontal and vertical distances (km and log(Pa)) between the observation and analysis grid point. $\rho_h$ and $\rho_v$ are tunable horizontal and vertical localization scales (km and log(Pa)). Subscripts $h$ and $v$ represent horizontal and vertical, respectively. This study set the vertical localization scale $\rho_v$ be 0.1 log(Pa) following Greybush et al. (2011) and Kondo et al. (2013). The horizontal localization scales of the LETKF were tuned manually before the experiments to minimize the first-guess root mean square error (RMSE) of the fourth layer (∼500 hPa) temperature of the SPEEDY-LETKF since this is among the most important variables for medium-range NWP.

The LETKF experiments used the adaptive multiplicative inflation method of Miyoshi (2011) in which the inflation factor $\beta$ was estimated adaptively. On the basis of sensitivity experiments for $\gamma$, we chose $\gamma = 1.5$. Sensitivity to $\gamma$ is discussed in section 4.3.3.

We first performed a one-year SPEEDY-LETKF experiment over the second year from January to December following the one-year spin-up. We then performed LETKF, LPF, and LPFGM experiments from January to April in the third year initialized by the first-guess ensemble of the LETKF at 0000 UTC on January 1 of the third year. The results from the last three months, i.e., February to April, were used for verification. We assessed the 6-h forecast or background RMSE for T

at the fourth model level (~500 hPa). While this study mainly discusses the RMSE and ensemble spread for T, similar results are observed for other variables. Here, this study will concentrate on the fourth-level T as an important variable for NWPs since the major goal of this study is to investigate the stabilities of LPF and LPFGM compared with the LETKF. Humidity verification or accounting for nonlinear observation operators and non-Gaussian observation errors are still crucial studies to investigate advantages of the LPF and LPFGM w.r.t. the LETKF.

## 4 Results and discussion

### 4.1 Experiments with a regularly distributed observing network

We first compare the LETKF, LPF, and LPFGM with REG2 in which the manually-tuned horizontal localization scale for the LETKF is 600 km. First, sensitivities to the horizontal localization scale ($\rho_h$) and RTPS parameter ($\alpha$) are investigated for the LPF. Figure 5 indicates the time-mean background RMSEs and ensemble spreads for LETKF and LPF. The LPF requires large inflation ($\alpha \geq 1.0$) to avoid filter divergence. With large inflation ($\alpha \geq 1.0$), the LPF experiment with $\rho_h = 600$ km resulted in the smallest RMSEs among the three LPF experiments. The best performing localization scale of LPF was found similar to that of the LETKF. The four LPF experiments exhibited larger RMSEs than the LETKF, as demonstrated by the previous studies with a low-dimensional L96 model (Poterjoy, 2016; Penny and Miyoshi, 2016; Farichi and Bouquet, 2018). The LPF experiment with $\rho_h = 700$ km shows filter divergence when $\alpha = 1.0$. The ensemble spreads of LPF are smaller than the RMSEs when $\alpha = 1.0$. This under dispersive ensemble is a typical behavior of LPF (e.g., Poterjoy and Anderson 2016).

Second, we compare the time series of the background RMSEs, ensemble spreads, and effective particle sizes $N_{eff}$ (Fig. 6). The RMSE and ensemble spread are consistent in the LETKF. However, the LPF shows generally smaller ensemble spreads than the RMSEs. Since the beginning of the experiments on January 1, the three LPF experiments showed rapid increases in the ensemble spread (Fig. 6 a), and a rapid decrease in the effective particle size (Fig. 6 b) within two weeks. After the rapid changes, the ensemble spreads and effective particle sizes were stabilized. The three LPF experiments increased RMSEs until the beginning of March. The LPF with $\rho_h = 500$ km showed filter divergence in April. The LPF with $\rho_h = 400$ km and $\rho_h = 600$ km also showed increasing trends in RMSE until the end of April while their ensemble spreads and effective particle sizes were stable.

Next, sensitivities to the RTPS parameter ($\alpha$), resampling frequency ($N_0$), and forgetting factor ($\tau$) are investigated for the LPFGM. The best performing localization scale of LPF was identical to that of the LETKF (Fig. 5). Therefore, we choose the same 600-km localization scale for the LPFGM. Figure 7 compares the time-mean background RMSEs and ensemble spreads for LETKF and LPFGM. In the LPFGM experiments, we investigate six experimental settings for $\tau = 0.0$, 0.1, 0.2, 0.3, 0.5, and 1.0. With $\tau = 0.0$, (Fig. 7 a), the LPFGM shows RMSEs similar to those of LETKF with best performing parameters for $\alpha$ and $N_0$. Regulating the resampling frequency ($N_0$) would be needed for LPFGM because the LPFGM shows filter divergence when resampling is employed at all assimilation steps excluding the case of $\alpha = 1.0$ (magenta line in Fig. 7

a). The LPFGM employs two ensemble updates: the Gaussian mixture and resampling. Owing to the first update by the Gaussian mixture, the LPFGM would not require resampling at all assimilation steps. With $\tau = 0.0$, the best performing relaxation parameter $\alpha$ is approximately 0.6–0.9, excluding the case employing resampling at all assimilation steps.

Increasing $\tau$ leads to the LPFGM being less sensitive to the resampling parameter $N_0$ and RTPS parameter $\alpha$ (Fig. 7 b-f), implying that the LPFGM is more stable when the LPFGM forgets weights ($\tau \geq 0.1$). Additionally, the LPFGM requires less inflation (i.e., smaller $\alpha$) when the forgetting factor $\tau$ becomes larger. Owing to the first update by the Gaussian mixture, succeeding weights completely may be unnecessary for the LPFGM. In addition, the method of succeeding weights in the *local* PF is not trivial because the weights would move with the dynamical flow. Because the results of the LPFGM experiments without weight succession ($\tau = 1.0$) were superior to those with weight succession ($\tau = 0.0$), the remainder of this paper focuses on the LPFGM experiments without weight succession ($\tau = 0.0$).

The time series of background RMSEs, and effective particle sizes for LPFGM experiments show that the LPFGM with every-time resampling ($N_0 = 40$) exhibits large RMSEs and the smallest effective particle sizes (Fig. 8, magenta). In contrast, the LPFGM experiments with infrequent resampling ($N_0 = 10$, 5, and 2) show small RMSEs similar to that of the LETKF and maintain larger effective particle sizes than the LPF (Fig. 8, red, blue and green lines).

Finally, we compare the spatial patterns of the RMSEs at a fourth model level for the LETKF, LPF, and LPFGM with best performing parameter settings (Fig. 9). The LETKF and LPFGM show larger RMSEs in the tropics and polar regions (Figs. 9 a and c), possibly because of uncertainties from convective dynamics in the tropics and sparser observations in the polar regions. The LPF shows a larger RMSE than the LETKF globally (Fig. 9 b, d). By contrast, slight improvements were observed globally in the LPFGM relative to the LETKF as indicated by generally warm colors in Fig. 9 (e), especially around the North Pole.

## 4.2 Experiments with a realistic observing network

Here we compare the LETKF, LPF, and LPFGM with the realistic observing network RAOB. With this spatially inhomogeneous observing network, all LPF experiments showed filter divergence, even with a broad range of the localization and RTPS parameters. Since there are fewer assimilated observations in RAOB than in REG2, the weight collapse would not be the primary cause of the LPF filter divergence. Because the LPF creates posterior particles by linearly combining prior particles, it is preferable for the LPF to have observations within the range of the prior particles. Filter divergence must be prevented by maintaining synchronization between the LPF and the observations. The LPF would require more observations for the synchronization than the LETKF, which was also shown in the authors' initial tests with L96. In addition, the LPFGM was unstable with weight succession ($\tau = 0.0$). Therefore, this section focuses on the comparison of the LETKF and LPFGM without weight succession ($\tau = 1.0$) only.

First, sensitivities to the RTPS parameter ($\alpha$) and resampling frequency ($N_0$) are investigated for the LPFGM. Figure 10 compares the time-mean background RMSEs and ensemble spreads for the LETKF and LPFGM. The best performing horizontal localization scale for the LETKF is 1100 km, and the same localization scale is used for the LPFGM. The LPFGM

with every-time resampling ($N_0 = 40$) shows the largest RMSEs (Fig. 10 a, magenta). By contrast, some LPFGM experiments show smaller RMSEs than the LETKF (Fig. 10 b). In this experimental setting, the resampling parameter of $N_0 = 2.0$ shows the most stable performance. With $N_0 = 2.0$, the LPFGM outperforms the LETKF slightly for $\alpha = 0.5$–$0.7$ (green line of Fig. 10 b).

The time series of the background RMSEs and effective particle sizes show that the LPFGM with every-time resampling ($N_0 = 40$; Fig. 11 a, magenta), and relatively-frequent resampling ($N_0 = 10$; Fig. 11 a, red) caused filter divergences. The times when the filters diverged seem to correspond to the times when the effective particle sizes decreased (Fig. 11 b, magenta and red). This reduction in effective particle size is a typical sign of filter divergence for PF (Snyder et al., 2008). One may speculate the assumption of $\widehat{\mathbf{R}}_t \approx \mathbf{R}_t$ in the LPFGM (section 2.3) would be a reason for the filter divergence. However, adopting the appropriate norm ($\widehat{\mathbf{R}}_t = \mathbf{R}_t + \mathbf{H}_t \widehat{\mathbf{P}}_t^b \mathbf{H}_t^T$) for the LPFGM did not eliminate the filter divergence (not shown). An effective solution for avoiding the filter divergence with every-time resampling is still unknown, but a possible reason is the method of creating a transform matrix using the Monte-Carlo method that approximates two ideal conditions (Eq. 19 and a spatially smooth transition). For deeper understanding, using other resampling techniques such as the optimal transport (see section 4.3.2) would be useful. In contrast, the LPFGM experiments with infrequent resampling ($N_0 = 5$ and $2$) show stable RMSEs and maintain a larger effective particle size. These stable LPFGM experiments maintain the amplitude of the ensemble spreads similar to that of the RMSEs.

We further explore regions where the LPFGM outperforms the LETKF. Figure 12 compares the space-based patterns of the background RMSEs for T (K) at the fourth model level for LETKF and LPFGM. Errors in sparsely recognized regions, such as the South Pacific Ocean, Indian Ocean, and polar regions, tend to be larger (Figs. 12 a and b). The LPFGM outperforms the LETKF in such sparsely observed regions (Fig. 12 c). The regions showing improvements correspond to the regions where the ensemble spreads and first-guess non-Gaussianity are larger (Fig. 13). Here, we measure the non-Gaussianity by Kullback–Leibler divergence (Kullback and Leibler, 1951).

The first update of the LPFGM (the Gaussian mixture) is similar to the LETKF update as discussed. Therefore, the improvements of the LPFGM with respect to the LETKF would be owing to the resampling step. Previous studies demonstrated that the LPF outperformed the LETKF for non-Gaussian DA with non-Gaussian observation and prior PDFs (Poterjoy et al. 2016a, 2016b, Penny and Miyoshi 2016). Therefore, the improvements of the LPFGM would come from the consideration of non-Gaussian prior PDF in sparsely observed regions.

Finally, we investigate the region where the difference in RMSE between the LETKF and LPFGM is large (120°W-60°W and 70°S-30°S; indicated by black dashed rectangles in Figs. 12 and 13). Figure 14 compares the time series of RMSE, ensemble spread, effective particle size, and Kullback–Leibler divergence averaged over the region for the LETKF and LPFGM. Here, we conducted an additional LPFGM experiment that employs no resampling (i.e., $N_0 = 0$) to investigate the importance of resampling. Vertical red lines in Fig. 14 represent the cases when the LETKF has large RMSEs greater than 0.8 K. Figure 14 (a) indicates that the LPFGM tends to mitigate large RMSEs in contrast to the LETKF. Since no significant

increase is seen in Kullback–Leibler divergence at the four cases (Fig. 14 c), the large RMSEs of LETKF would not be caused by analyses with highly non-Gaussian first-guess ensemble. For the last three cases, the LPFGM has significantly smaller RMSEs than the LETKF. The improvement of the LPFGM would be partially led by larger ensemble spread than the LETKF (Fig. 14 b). However, the LPFGM with resampling ($N_0 = 2$) outperforms the LPFGM without resampling ($N_0 = 0$), indicating that resampling improves the RMSE. The LETKF and LPFGM exhibit reduced effective particle size at the three later cases (Fig. 14 d). Because of the smaller effective particle size, fewer ensembles predict states that are closer to observations, and the LPFGM employed more resampling in the region. In addition to the particle shift, the LPFGM would reduce RMSE further by the resampling.

The LPFGM occasionally resulted in larger RMSE than the LETKF (e.g., beginning of March in Fig. 14a). However, the LPFGM has the potential to outperform the LETKF overall by reducing the RMSE in sparsely observed regions. While operational NWP systems assimilate massive observations, the LPFGM would be useful when spatially and temporally sparse observations are used, e.g., in the twentieth century reanalysis projects (e.g. Compo et al. 2011; Laloyaux et al. 2018) and paleo climate reconstructions (Acevedo et al. 2017; Okazaki and Yoshimura 2017).

### 4.3 Factors requiring further investigation

Finally, we discuss other issues for potential further improvement of the LPF and LPFGM in future studies.

### 4.3.1 Inflation

This study used the RTPS to inflate the posterior perturbation for the LPF and LPFGM. As shown in Figs. 5, 7 and 10, the RMSEs of the LPF and LPFGM are sensitive to the RTPS parameter. There is a need to investigate methods that estimate the RTPS parameter adaptively, as in the EnKF (Ying and Zhang, 2015; Kotsuki et al., 2017b). However, the adaptive relaxation methods used in the EnKF cannot be applied directly to the LPF because they use the innovation statistics (Desroziers et al., 2005) that assume analysis updates by the Kalman gain. Therefore, substantially different adaptive relaxation methods should be investigated for the LPF and LPFGM.

An alternative method of inflation is adding random noise to the transform matrix (Potthast et al., 2019). Regulating the amplitude of the random noise was not trivial in the authors' preliminary experiments with L96 (not shown). Too small random noise results in loss of diversity of posterior particles, whereas excessively large random noise results in an overly dispersive ensemble. Potthast et al. (2019) also suggested estimating the amplitudes of the random noise based on the innovation statistics. However, determining the minimum and maximum values of the amplitudes was not trivial from the authors' experience with L96. Therefore, a method to determine the optimum amplitude of random noise should be investigated as well. Moreover, additive inflation methods, as used in the EnKF, can be beneficial for producing diverse prior particles (Mitchell and Houtekamer, 2000; Corazza et al., 2003). For example, Penny and Miyoshi (2016)'s LPF applied an additive

inflation Gaussian noise whose variance is scaled to a magnitude of local analysis error variance. Investigating better and adaptive inflation methods is very important to stabilize LPFs.

### 4.3.2 Transform matrix for LPF

The transform matrix significantly affects the filter performance. Farchi and Bocquet (2018) demonstrated that the optimal transport method of Reich (2013) resulted in lower RMSE than the commonly used stochastic uniform resampling approach with the L96 model. However, it is not clear if optimal transport is beneficial for high-dimensional systems such as the SPEEDY model.

In this study, all-time resampling was detrimental for the LPFGM experiments (Figs. 7 and 10). This may be due to the MC resampling that satisfies Eq. (19) ($\bar{\mathbf{x}}_t^b = \sum_{i=1}^m \mathbf{x}_t^{b(i)} \cdot w_t^{a(i)}$) not exactly, but approximately. With a small effective particle size, the MC resampling almost satisfies Eq. (19) since the resampling is not affected by the sampling noise in the selection of particles. In contrast, satisfying Eq. (19) is more difficult when the effective particle size is larger due to the sampling noise. A possible solution to this problem is to use the optimal transport method (Reich 2013) that satisfies Eq. (19) exactly. Another essential property for local PFs is the spatially smooth transition of the transform matrix. The optimal transport method would also be useful for ensuring the spatially smooth transition of the transform matrix because the posterior weights in nearby grids are typically similar. It is necessary to investigate better methods for generating resampling matrices.

### 4.3.3 Tunable parameters of LPF and LPFGM

The tunable parameters of the LPF and LPFGM should be investigated further. For example, parameter $N_0$, which controls the resampling frequency, significantly affected the filter accuracy and stability. Adaptive determination of this parameter can prevent time-consuming parameter tuning.

The sensitivity to parameter $\gamma$ should also be investigated for the LPFGM. WP22 proposed using $\gamma = 2.5$. Since $\gamma$ controls the amplitude and width of Gaussian kernels, the filter accuracy and stability of the LPFGM would be sensitive to this parameter. We briefly examined the sensitivity to this parameter with REG2 and RAOB (Fig. 15; $\gamma = 2.5$, 1.5, and 0.5 for red, blue, and green lines). The results indicate that the filter accuracy of the LPFGM is sensitive to this parameter, especially for spatially-inhomogeneous RAOB observing network. For our experimental settings, the best performing $\gamma$ was 1.5 for both REG2 and RAOB. However, the optimal parameter would differ for different models, observing networks, and observing frequency. Also, there might be a relation between the optimal $\gamma$ and ensemble size. Therefore, adaptive determination of this parameter is helpful.

The LPFGM without weight succession ($\tau = 1.0$) resulted in lower RMSE than that with weight succession ($\tau = 0.0$). However, the optimal parameter $\tau$ would be somewhere between 0.0 and 1.0. In addition, the weight succession at each model grid point would be suboptimal. Hence, the method for weight succession should also be explored further.

**5 Summary**

This study aims to develop a software platform for the LETKF, LPF, and LPFGM with the intermediate global atmospheric model SPEEDY. The main results of this investigation are briefly listed as follows:

1) The LPF and LPFGM were developed by only minor modifications to the existing LETKF system with the SPEEDY model.

2) With dense observations (REG2), the LPF showed stable filter performance with large inflation by the RTPS. The best performing localization scale of the LPF was identical to that of the LETKF. The LPF forecast was less accurate than the LETKF forecast. With sparse observations (RAOB), the LPF did not work.

3) The LPFGM showed more stability and lower forecast RMSEs than the LPF. In addition to the RTPS parameter, regulating the resampling frequency and the amplitude of Gaussian kernels was important for the LPFGM. The LPFGM without weight succession resulted in more stability and lower RMSEs than that with weight succession. With RAOB, the LPFGM forecast was more accurate than the LETKF forecast in sparsely observed regions where the background ensemble spread and non-Gaussianity are larger.

As discussed in section 4.3, there is much room for improvement in the LPF and LPFGM. While the LPFGM potentially provides a more accurate forecast than the LETKF, the LPFGM has more tunable parameters than the LETKF. Manually tuning these parameters by trying numerous experiments is computationally expensive. Therefore, adaptive methods for determining such tuneable parameters need to be explored. Also, it is important to investigate computationally more efficient methods to generate resampling matrices for the LPF and LPFGM.

The SPEEDY-based LETKF, LPF, and LPFGM used in this study are available as open-source software on Github (https://github.com/skotsuki/speedy-lpf). This can act as a useful platform to investigate the LPF and LPFGM further in comparison with the well-known LETKF.

**Appendix A: Derivations of Kalman gain and analysis error covariance**

Here we describe derivations of Kalman gain and analysis error covariance (Eqs. 5 and 6). The Kalman gain $\mathbf{K}_t = \mathbf{P}_t^b \mathbf{H}_t^T (\mathbf{H}_t \mathbf{P}_t^b \mathbf{H}_t^T + \mathbf{R}_t)^{-1}$ can be changed as follows:

$$\mathbf{K}_t = [(\mathbf{P}_t^b)^{-1} + \mathbf{H}_t^T \mathbf{R}_t^{-1} \mathbf{H}_t]^{-1}[(\mathbf{P}_t^b)^{-1} + \mathbf{H}_t^T \mathbf{R}_t^{-1} \mathbf{H}_t] \mathbf{P}_t^b \mathbf{H}_t^T (\mathbf{H}_t \mathbf{P}_t^b \mathbf{H}_t^T + \mathbf{R}_t)^{-1} \tag{A1}$$

$$= [(\mathbf{P}_t^b)^{-1} + \mathbf{H}_t^T \mathbf{R}_t^{-1} \mathbf{H}_t]^{-1}[\mathbf{H}_t^T + \mathbf{H}_t^T \mathbf{R}_t^{-1} \mathbf{H}_t \mathbf{P}_t^b \mathbf{H}_t^T](\mathbf{H}_t \mathbf{P}_t^b \mathbf{H}_t^T + \mathbf{R}_t)^{-1} \tag{A2}$$

$$= [(\mathbf{P}_t^b)^{-1} + \mathbf{H}_t^T \mathbf{R}_t^{-1} \mathbf{H}_t]^{-1} \mathbf{H}_t^T \mathbf{R}_t^{-1}[\mathbf{R}_t + \mathbf{H}_t \mathbf{P}_t^b \mathbf{H}_t^T](\mathbf{H}_t \mathbf{P}_t^b \mathbf{H}_t^T + \mathbf{R}_t)^{-1} \tag{A3}$$

$$= [(\mathbf{P}_t^b)^{-1} + \mathbf{H}_t^T \mathbf{R}_t^{-1} \mathbf{H}_t]^{-1} \mathbf{H}_t^T \mathbf{R}_t^{-1}. \tag{A4}$$

Using $\mathbf{K}_t = [(\mathbf{P}_t^b)^{-1} + \mathbf{H}_t^T \mathbf{R}_t^{-1} \mathbf{H}_t]^{-1} \mathbf{H}_t^T \mathbf{R}_t^{-1}$, analysis error covariance $\mathbf{P}_t^a = (\mathbf{I} - \mathbf{K}_t \mathbf{H}_t) \mathbf{P}_t^b$ can be changed as follows:

$$\mathbf{P}_t^a = (\mathbf{I} - \mathbf{K}_t \mathbf{H}_t) \mathbf{P}_t^b = \mathbf{P}_t^b - \mathbf{K}_t \mathbf{H}_t \mathbf{P}_t^b \tag{A5}$$

$$= \mathbf{P}_t^b - [(\mathbf{P}_t^b)^{-1} + \mathbf{H}_t^T \mathbf{R}_t^{-1} \mathbf{H}_t]^{-1} \mathbf{H}_t^T \mathbf{R}_t^{-1} \mathbf{H}_t \mathbf{P}_t^b \tag{A6}$$

$$= \mathbf{P}_t^b - [(\mathbf{P}_t^b)^{-1} + \mathbf{H}_t^T \mathbf{R}_t^{-1} \mathbf{H}_t]^{-1}[((\mathbf{P}_t^b)^{-1} + \mathbf{H}_t^T \mathbf{R}_t^{-1} \mathbf{H}_t) - (\mathbf{P}_t^b)^{-1}] \mathbf{P}_t^b \tag{A7}$$

$$= \mathbf{P}_t^b - \{\mathbf{I} - [(\mathbf{P}_t^b)^{-1} + \mathbf{H}_t^T\mathbf{R}_t^{-1}\mathbf{H}_t]^{-1}(\mathbf{P}_t^b)^{-1}\}\mathbf{P}_t^b \tag{A8}$$

$$= [(\mathbf{P}_t^b)^{-1} + \mathbf{H}_t^T\mathbf{R}_t^{-1}\mathbf{H}_t]^{-1}. \tag{A9}$$

Consequently, we can derivate $(\mathbf{P}_t^a)^{-1} = (\mathbf{P}_t^b)^{-1} + \mathbf{H}_t^T\mathbf{R}_t^{-1}\mathbf{H}_t$ from Eq. (A9), and $\mathbf{K}_t = \mathbf{P}_t^a\mathbf{H}_t^T\mathbf{R}_t^{-1}$ from Eq. (A4).

**Appendix B: The number of required samples for the MC resampling**

This appendix investigates the number of required samples for the MC resampling to obtain accurate transform matrices stably. Here we assume that 10,000 samples are sufficient to obtain accurate transform matrix ($\mathbf{T}_{N_{MC}=10,000}$). An absolute error (AE) of transform matrix with $G$ samples ($\mathbf{T}_{N_{MC}=G}$) is defined to measure accuracy of the transform matrix:

$$AE = \sum_{i=1}^{m}\sum_{j=1}^{m}|\mathbf{T}_{N_{MC}=G}^{(i,j)} - \mathbf{T}_{N_{MC}=10,000}^{(i,j)}|. \tag{B1}$$

Figure B1 shows the AE as a function of the number of samples for ensemble members 10, 20, 40 and 80. For this investigation, we obtained 1,000 independent weights generated by uniform random numbers, and average, minimum and maximum AEs of 1,000 cases are shown. For four ensembles, AEs decrease rapidly for the first 1,000 samples, followed by gradual decrease until AE=0. To reach to 10 % of initial error at $N_{MC} = 1$, MC resampling requires about 70, 300, 400, and 600 samples for

ensemble members 10, 20, 40, and 80. It suggests that more ensemble requires more samples to obtain accurate transform matrix stably by the MC resampling.

**Author contributions**

SK developed the data assimilation system, conducted the experiments, analyzed the results and wrote this manuscript. TM is the PI and directed the research with substantial contribution to the development of this paper. KK developed the data

assimilation system with SK. RP contributed substantially to the introduction of the LPFGM to the standard local particle filter.

**Data and code availability**

Developed data assimilation system is available at (https://github.com/skotsuki/speedy-lpf), which is based on the exiting SPEEDY-LETKF system (https://github.com/takemasa-miyoshi/letkf). The data assimilation system used in this manuscript is archived on Zonodo (https://zenodo.org/record/6586309; doi: 10.5281/zenodo.6586309). Due to the large

volume of data (> 6 TB) and limited disk space, processed data and scripts for visualization are also shared on Zenodo at the same link.

**Competing interests**

The authors have no competing interests to declare.

## Acknowledgements

The authors thank the members of the Data Assimilation Research Team, RIKEN Center for Computational Science (R-CCS), for valuable suggestions and discussions. SK thanks Mr. Ken Oishi of Chiba University for useful discussions. This study was partly supported by the Japan Aerospace Exploration Agency (JAXA) Precipitation Measuring Mission (PMM), advancement of meteorological and global environmental predictions utilizing observational 'Big Data' of the social and scientific priority issues (Theme 4) to be tackled by using post K computer of the FLAGSHIP2020 Project of the Ministry of Education, Culture, Sports, Science and Technology Japan (MEXT), the Program for Promoting Researches on the Supercomputer Fugaku" (Large Ensemble Atmospheric and Environmental Prediction for Disaster Prevention and Mitigation) of MEXT JPMXP1020200305, the Initiative for Excellent Young Researchers of MEXT, JST AIP Grant Number JPMJCR19U2, the Japan Society for the Promotion of Science (JSPS) KAKENHI grants JP18H01549, 21H04571 and 21H05002, JST PRESTO MJPR1924, and IAAR Research Support Program of Chiba University. The experiments were performed using the Supercomputer for earth Observation, Rockets, and Aeronautics (SORA) at JAXA.

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

## Algorithms

---

**Algorithm 1:** Generation of the resampling matrix **T**

---

**Requirement: $\mathbf{r}_{sorted}$ and $\mathbf{w}_{t,acc}$**

| | |
|---|---|
| 1: | All components of **T** and **z** are initialized to be 0 |
| | #z: an indicator of resampled particles |
| 2: | **do** $j$=1 to $m$                # index of posterior particle |
| 3: | **do** $i$=1 to $m$                # index of prior particle |
| 4: | **if** $\{w_{t,acc}^{(i-1)} < r_{sored}^{(j)} \le w_{t,acc}^{(i)}\}$ $z^{(j)} = i$   # resampled particle |
| 5: | **end do** |
| 6: | **end do** |
| 7: | **do** $j$=1 to $m$                # (loop for posterior particle) |
| 8: | $k = z^{(j)}$ |
| 9: | **if** $\{\mathbf{T}^{(k,k)} = 0\}$ **then** |
| 10: | $\mathbf{T}^{(k,k)}$ =1              # diagonal components |
| 11: | $z^{(j)} = 0$ |
| 12: | **end if** |
| 13: | **end do** |
| 14: | **do** $j$=1 to $m$               # (loop for posterior particle) |
| 15: | $k = z^{(j)}$ |
| 16: | **if** $\{k \ne 0\}$ **then** |
| 17: | **do** $l$=1 to $m$            # (loop for columns of **T**) |
| 18: | **if** $\{sum(\mathbf{T}^{(:,l)}) = 0\}$ **then** |
| 19: | $\mathbf{T}^{(k,l)}$ =1          # off-diagonal components |
| 20: | $z^{(j)} = 0$ |
| 21: | exit |
| 22: | **end if** |
| 23: | **end do** |
| 24: | **end if** |
| 25: | **end do** |

**Tables**

Table 1: Computational complexities of LETKF, LPF, and LPFGM. Each step corresponds to the steps in Fig. 3. Cross-marks represent the steps used for LETKF, LPF and LPFGM.

| | Step | Computational complexity | LETKF | LPF | LPFGM |
|---|---|---|---|---|---|
| $C_\mathrm{H}$ | A | $m$ applications of $H$ | X | X | X |
| | B | $2mp$ | X | X | X |
| | C | $2nm$ | X | X | X |
| $C_\mathrm{L}$ | D | ※1 | X | X | X |
| | E | $mp_L$ (※2) | X | | X |
| | F | $\leq 2m^2 p_L$ | X | | X |
| | G | $O(m^3)$ | X | | X |
| | H | $O(m(m+p_L))$ | X | | |
| | I | $O(m(m+p_L))$ | | | X |
| | J | $O(mp_L)$ (※2) | | X | X |
| | K | $O(m^2 N_{MC})$ | | X | X |
| | L | $\leq 2m^3$ | | | X |
| $C_\mathrm{H}$ | M | $\leq 2nm^2$ | X | X | X |
| | N | $O(nm)$ | X | X | X |

※1: This computation depends on the localization scale

※2: These computations assume the diagonal **R** (i.e., uncorrelated observation error)

$n$: system size

$m$: ensemble size

$p$: the number of observations

$p_L$: the number of local observations

$N_{MC}$: the number of times for generating resampling matrices by **Algorithm 1**

**Table 2**: List of experiments

| Obs | DA | Purpose | $\rho_h$ (km) | $\alpha$ | $\gamma$ | $N_0$ | $\tau$ | Figures |
|---|---|---|---|---|---|---|---|---|
| | LETKF | control experiment | 600 | ※1 | / | / | / | |
| | LPF | sensitivity to $\rho_h$ and $\alpha$ | 400/500/ 600/700 | 0.0-1.0 | / | / | / | Figs. 5, 6, 9 |
| REG2 | LPFGM | sensitivity to $\alpha$, $N_0$, and $\tau$ | 600 | 0.0-1.0 | 1.5 | 2/4/ 10/40 | 0.0/0.1/ 0.2/0.3/ 0.5/1.0 | Figs. 7, 8, 9 |
| | LPFGM | sensitivity to $\alpha$ and $\gamma$ | 600 | 0.0-1.0 | 0.5/1.5/2.5 | 2 | 1.0 | Figs. 15 (a) |
| | LETKF | control experiment | 1,100 | ※1 | | | | |
| | LPF | sensitivity to $\rho_h$ and $\alpha$ | 1,100 | 0.0-1.0 | / | / | / | not shown |
| RAOB | LPFGM | sensitivity to $\alpha$ and $N_0$ | 1,100 | 0.0-1.0 | 1.5 | 2/4/ 10/40 | 1.0 | Figs. 10, 11, 12, 13 |
| | LPFGM | sensitivity to $N_0$ | 1,100 | 0.60 | 1.5 | 0/2 | 1.0 | Fig. 14 |
| | LPFGM | sensitivity to $\alpha$ and $\gamma$ | 1,100 | 0.0-1.0 | 0.5/1.5/2.5 | 2 | 1.0 | Figs. 15 (b) |

$\rho_h$: horizontal localization scale (Eq. 38)

$\alpha$: relaxation parameter of RTPS (Eq. 8)

 $\gamma$: amplitude of Gaussian kernel of the LPFGM (Eq. 25)

$N_0$: parameter controls resampling frequency (Eq. 22)

$\tau$: forgetting factor of weight (Eq. 23)

※1: Instead of RTPS, the LETKF uses Miyoshi (2011)'s adaptive multiplicative inflation

**Figures**

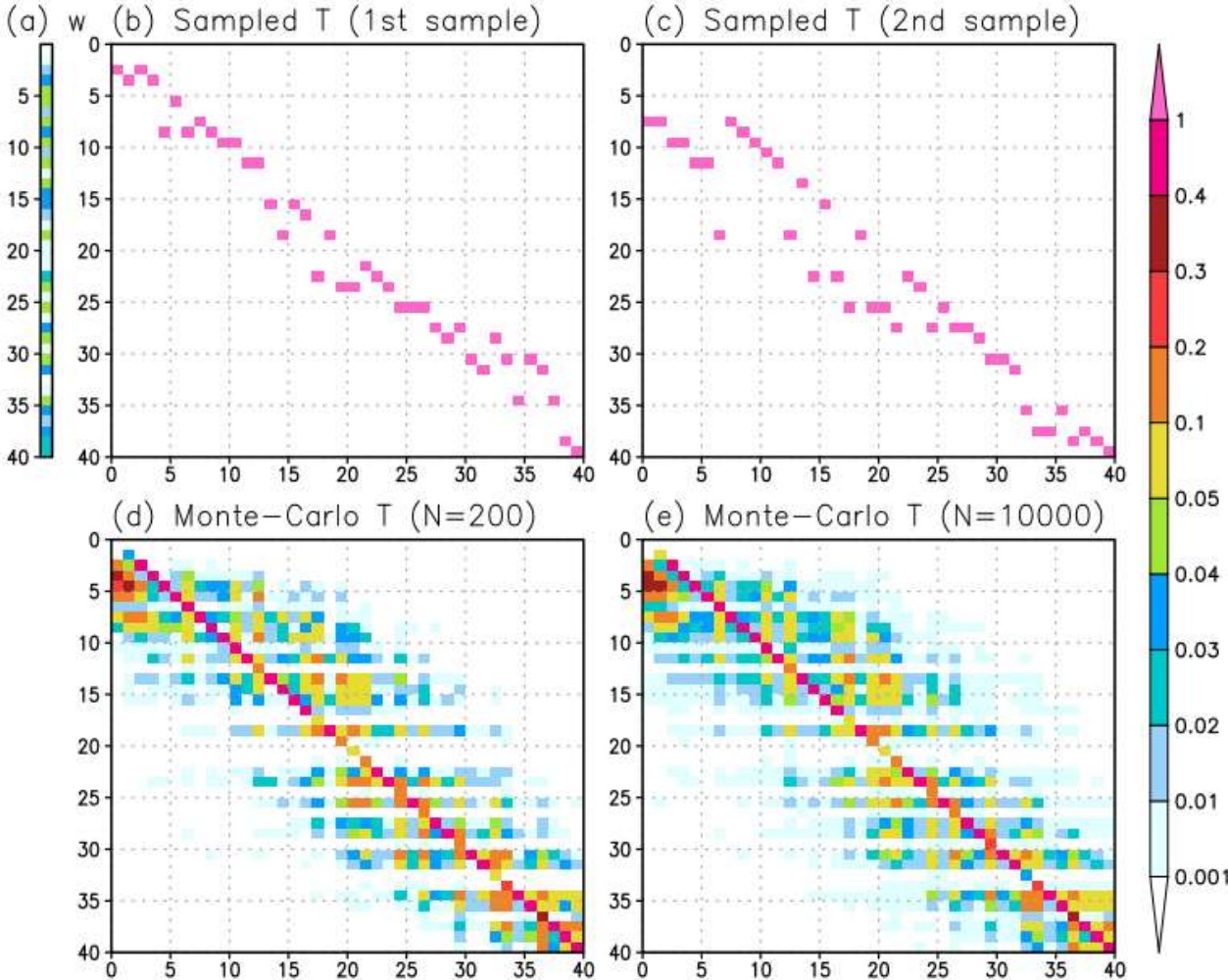

**Figure 1:** Examples of the resampling matrix in the case of 40 particles with a given weight generated by uniform random numbers. (a) Weight for 40 particles. (b, c) Examples of sampled resampling matrices by **Algorithm 1** using different random numbers **r**. (d, e) Resampling matrices using the Monte Carlo stochastic approach with 200 and 10,000 sampled matrices, respectively.

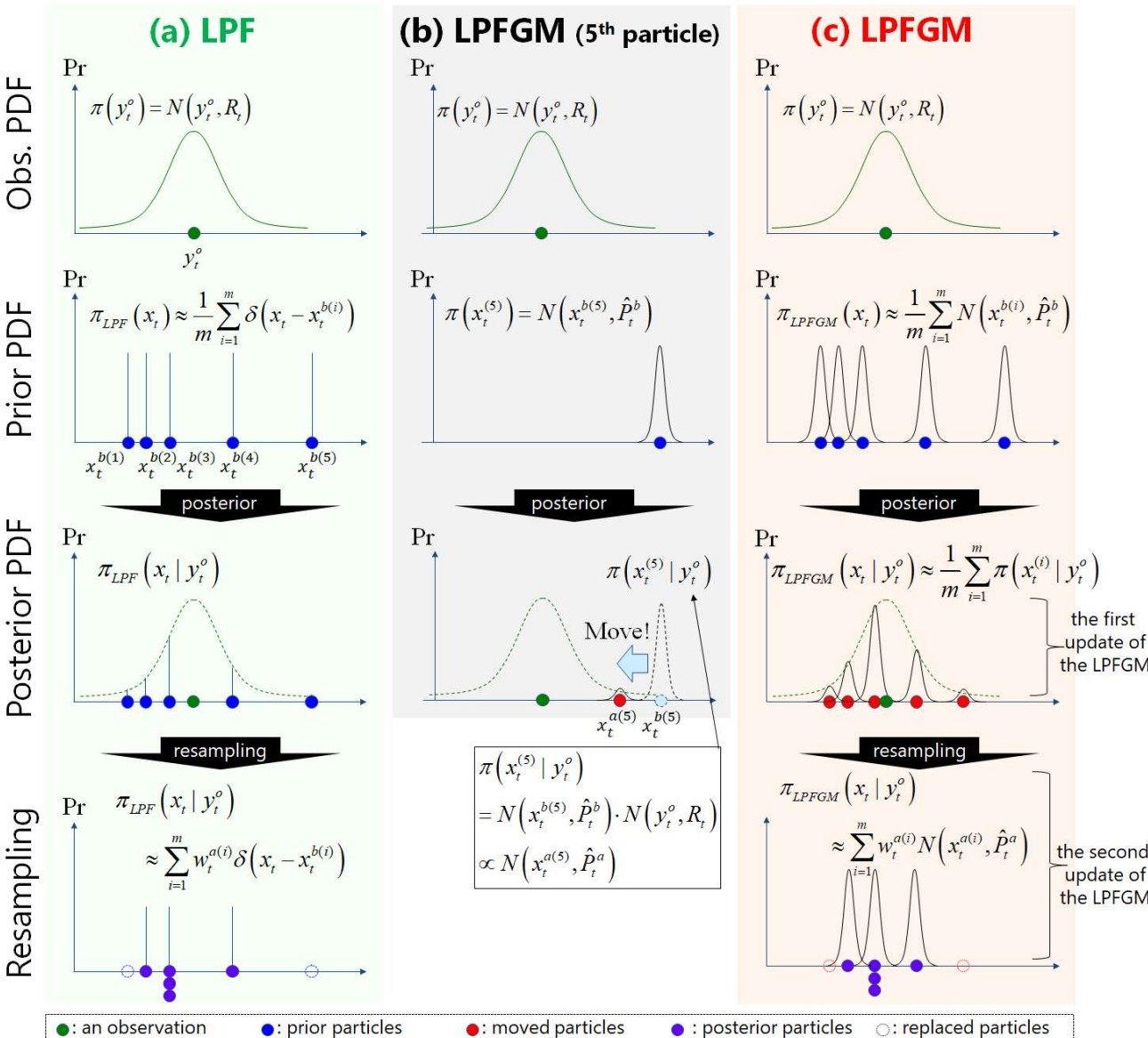

**Figure 2:** A simple scalar example of observation, prior and posterior PDFs for (a) LPF, (b) 5th particle of LPFGM and (c) LPFGM. Green, blue, red, purple, and dashed circles represent an observation and prior particles, moved particles, posterior particles, and replaced particles respectively.

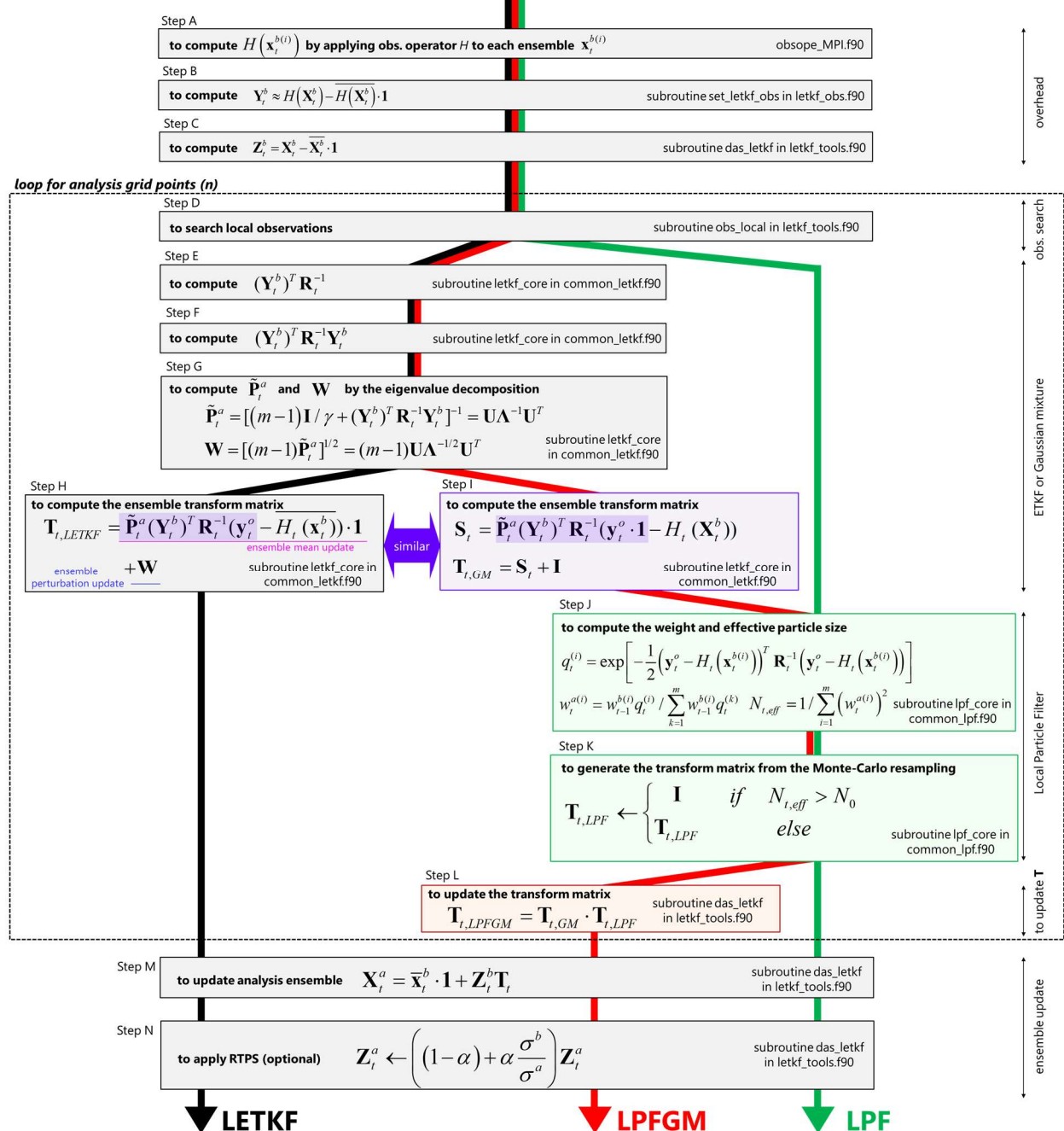

**Figure 3:** Workflows of LETKF, LPFGM, and LPF. Gray, purple, green, and red boxes represent the components of LETKF, a similar component as in the LETKF, the components of the LPF, and a unique component of the LPFGM, respectively. Subroutines and source files are also described in boxes.

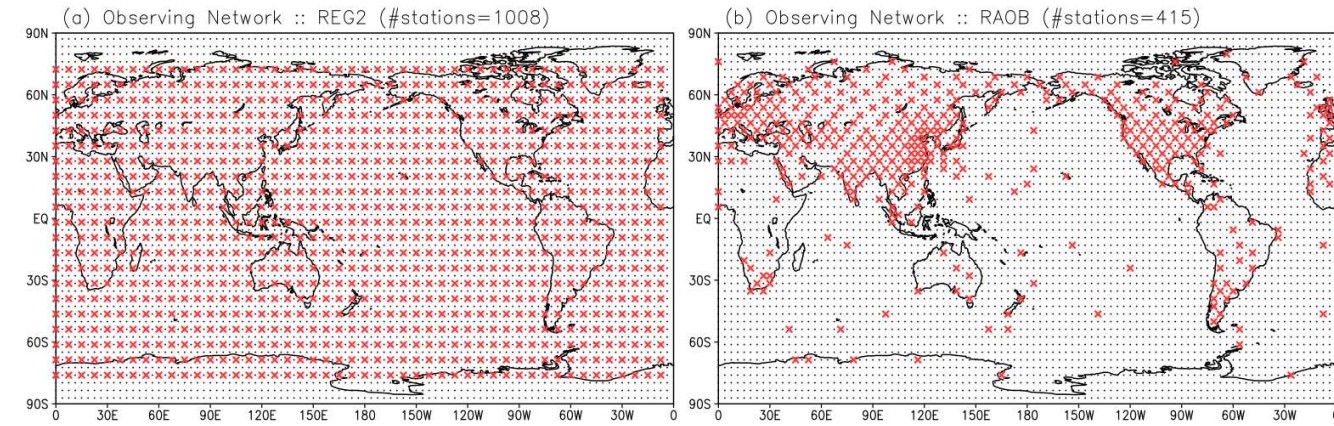

**Figure 4:** Observing networks for (a) REG2 and (c) RAOB experiments. Small black dots and red crosses represent model grid points and observing points, respectively.

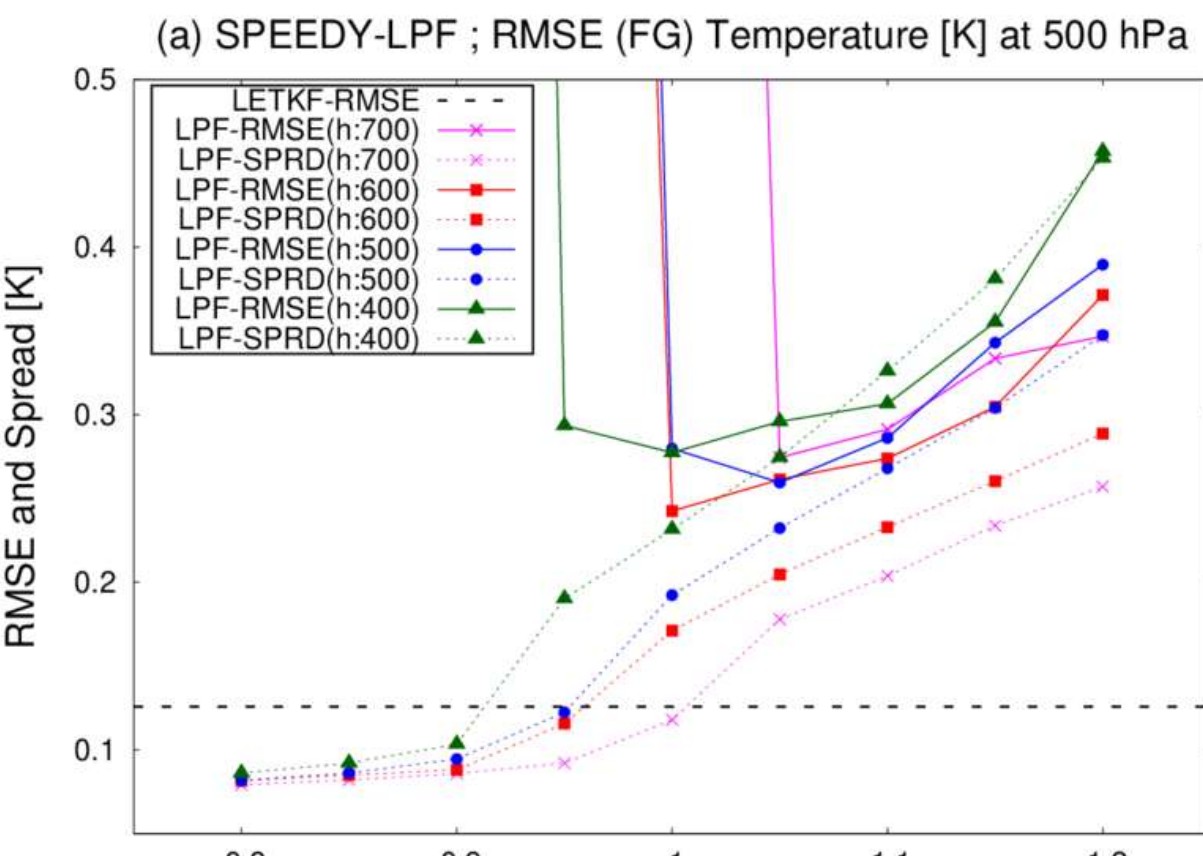

**Figure 5:** Time-mean background RMSEs (solid lines) and ensemble spreads (dashed lines) for T (K) at the fourth model level (~500 hPa) as a function of RTPS parameter $\alpha$ averaged over three months of the third year (February-April) with REG2 observations. Magenta, red, blue, and green lines are LPF experiments with horizontal localization scales of 700, 600, 500, and 400 km, respectively. Dashed lines represent RMSE of LETKF (0.1257 K) with adaptive multiplicative inflation instead of RTPS.

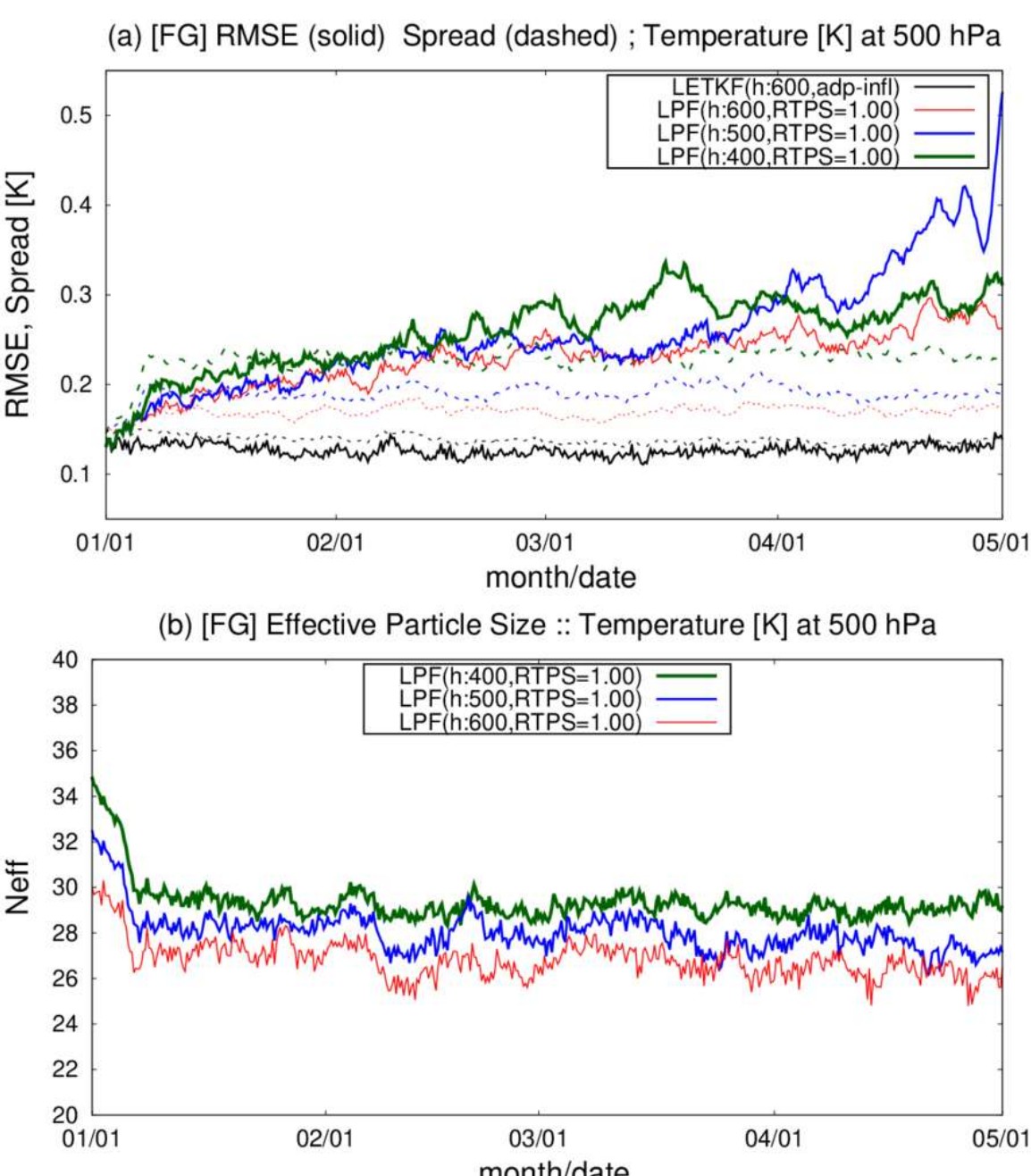

**Figure 6:** Time series of globally-averaged background (a) RMSEs (solid lines) and ensemble spreads (dashed lines) for T (K) and (b) effective particle size $N_{eff}$ at the fourth model level (~500 hPa) with REG2 observations. Black lines show the LETKF. Red, blue, and green lines are the LPF experiments with localization scales of 600, 500, and 400 km, respectively. RTPS parameter $\alpha$ is set to 1.00 in all three LPF experiments. The abscissa shows month/day.

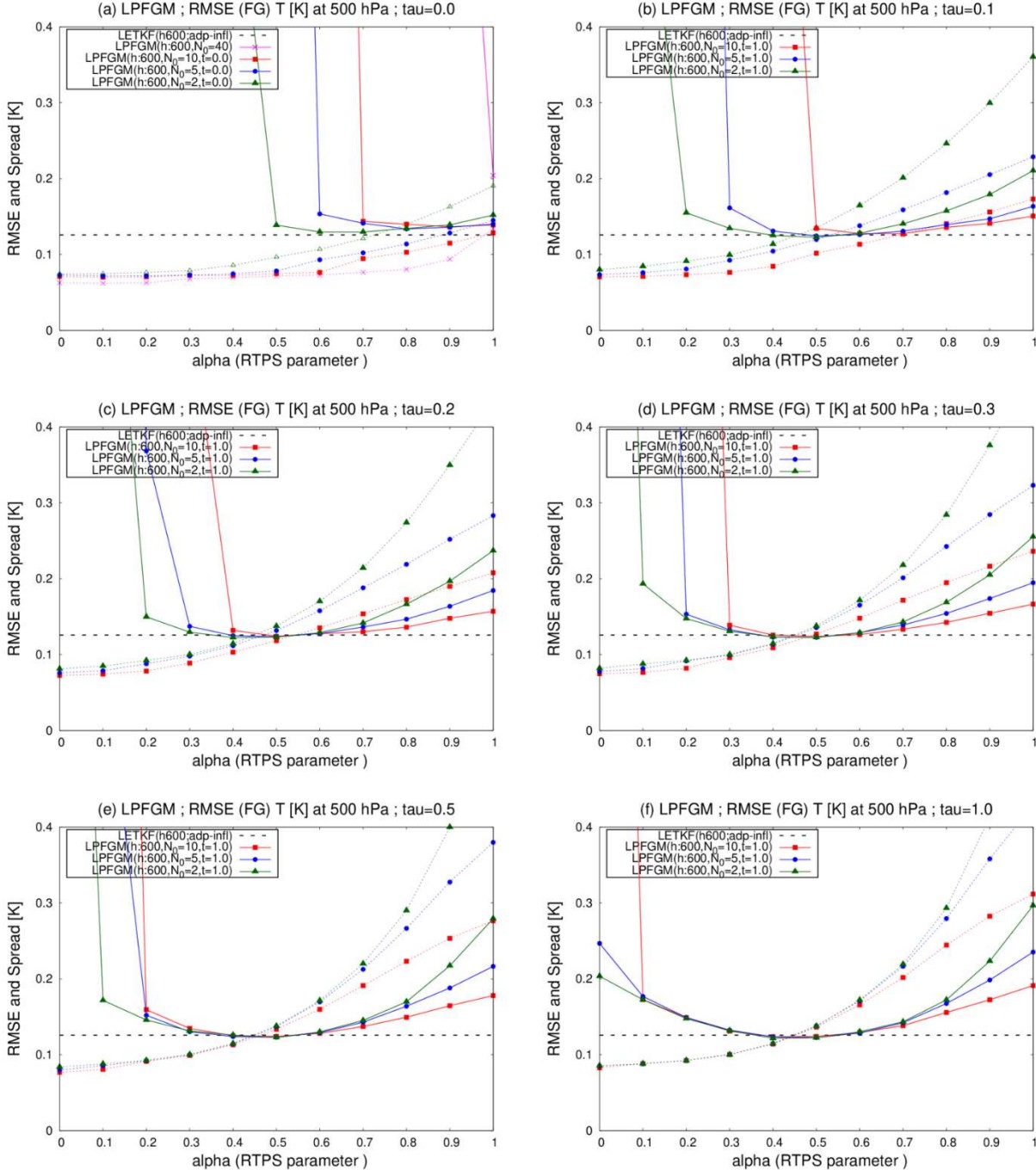

**Figure 7:** Similar to Figure 5, but for the LPFGM with REG2 and 600-km localization scale. Magenta, red, blue, and green lines show the cases with $N_0$ = 40, 10, 5, and 2, respectively. Dashed lines represent RMSE of LETKF (0.1257 K). Panels show the LPFGM with forgetting factors for (a) $\tau$ = 0.0, (b) $\tau$ = 0.1, (c) $\tau$ = 0.2, (d) $\tau$ = 0.3, (e) $\tau$ = 0.5 and (f) $\tau$ = 1.0, respectively.

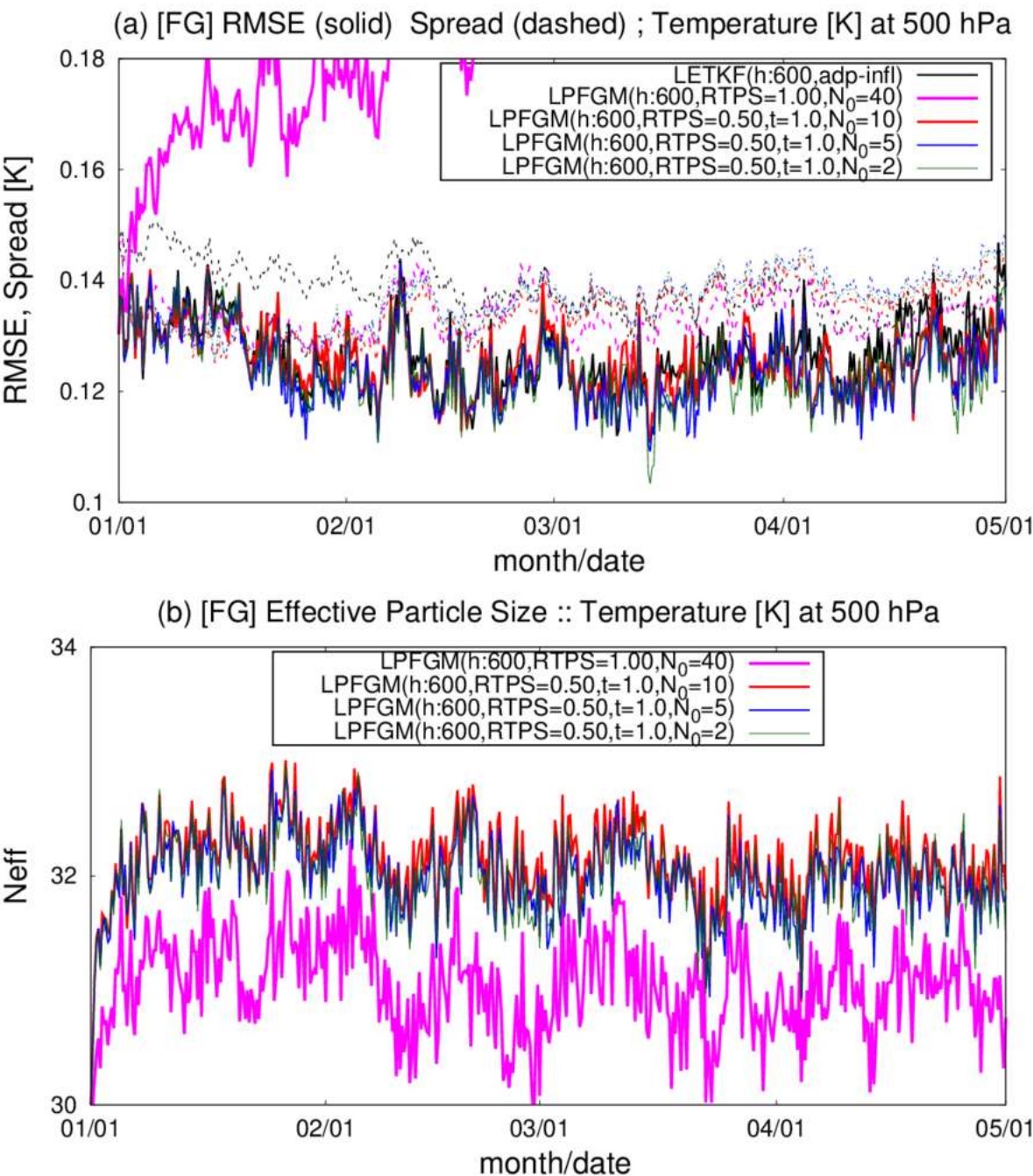

**Figure 8:** Similar to Figure 6, but showing the LPFGM experiments with REG2 and 600-km localization scale. Magenta, red, blue, and green lines are LPFGM experiments with $N_0$ = 40, 10, 5, and 2, respectively. Forgetting factor $\tau$ is 1.0. The RTPS parameter is set to 1.00 in the experiment with $N_0$ = 40 (i.e., all-time resampling) and to 0.50 for in other experiments.

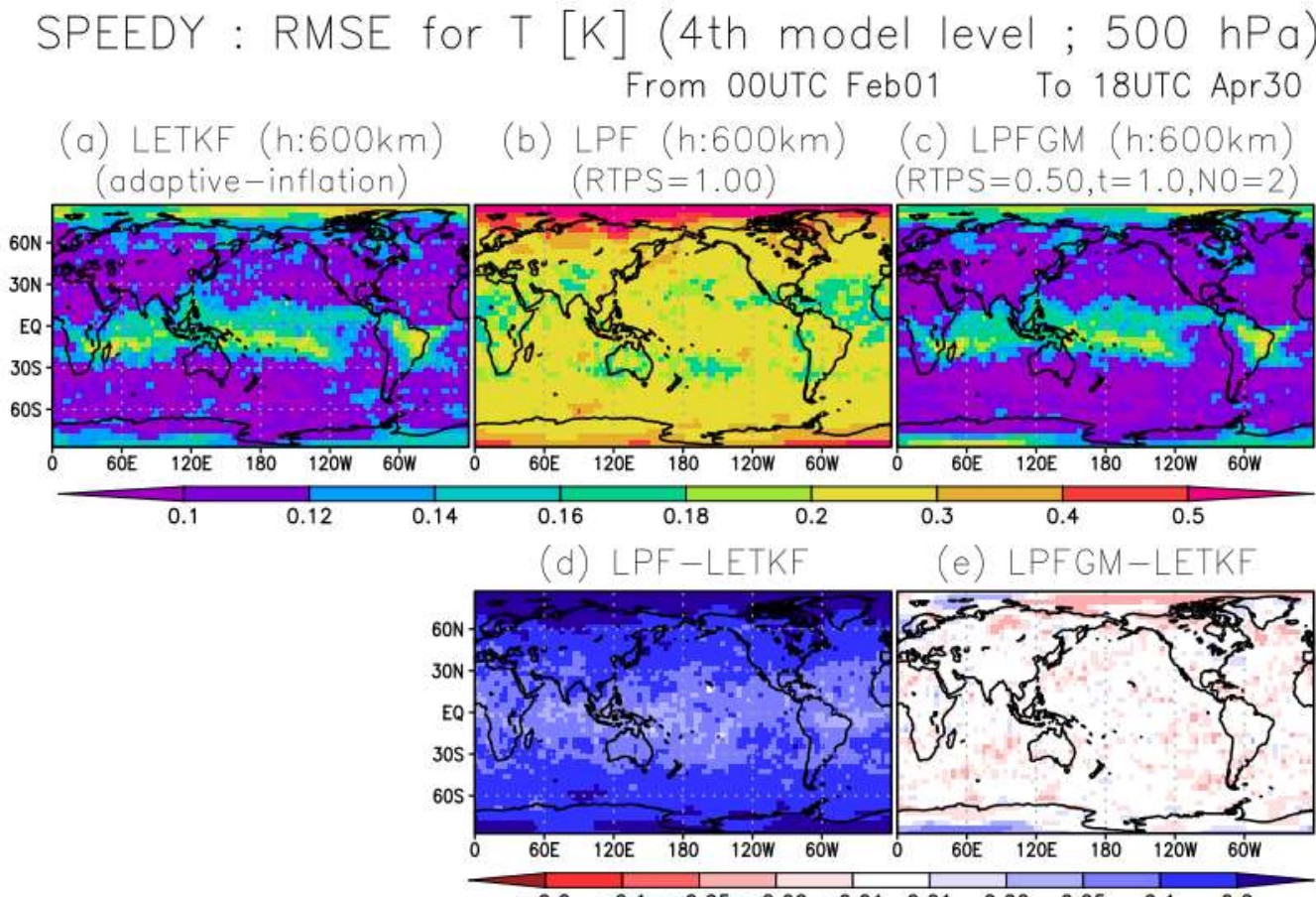

**Figure 9:** Spatial patterns of background RMSE for T (K) at the fourth model level (∼500 hPa) for (a) LETKF, (b) LPF, and (c) LPFGM with best performing localization scale and RTPS parameter, averaged over February-April with REG2 and 600-km localization scale. Panels (d) and (e) show the differences LETKF–LPF and LETKF–LPFGM, respectively. The warm

 (cold) color indicates that the LPF or LPFGM is better (worse) than the LETKF.

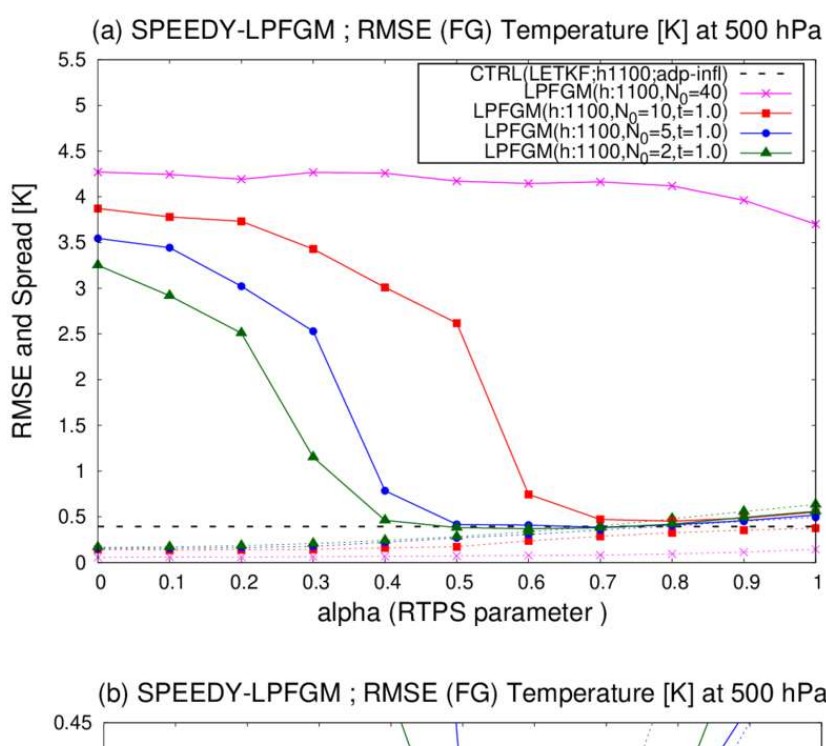

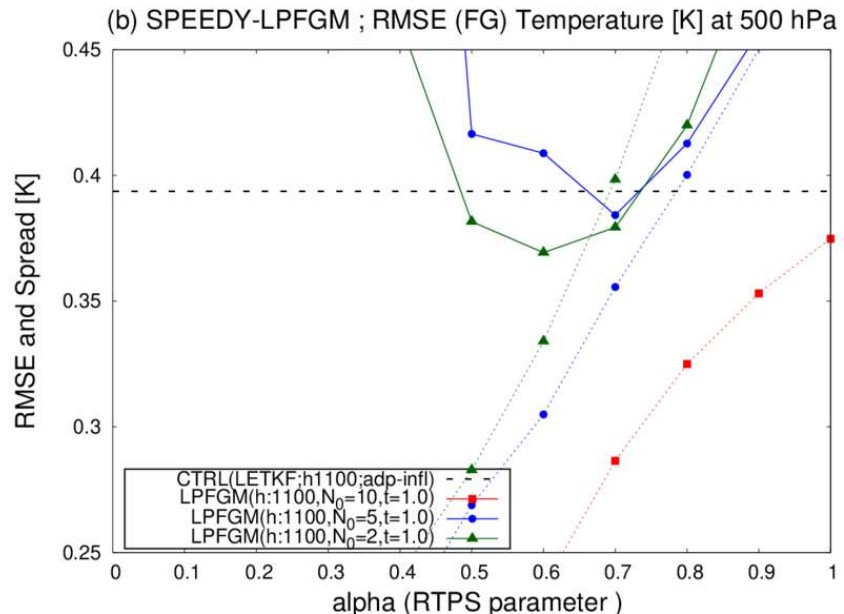

**Figure 10:** Similar to Figure 5, but showing the LPFGM experiments with the RAOB and 1100-km localization scale. Magenta, red, blue, and green lines are the LPFGM experiments with $N_0 = 40$, 10, 5, and 2, respectively. Forgetting factor $\tau$ is fixed at 1.0. Dashed black lines represent the RMSE of the LETKF (0.4762 K). (b) enlarges (a) for the range of the RMSE of 0.25–0.45.

(a) [FG] RMSE (solid)  Spread (dashed) ; Temperature [K] at 500 hPa

LETKF(h:1100,adp-infl)
LPFGM(h:1100,RTPS=1.00,$N_0$=40)
LPFGM(h:1100,RTPS=0.60,t=1.0,$N_0$=10)
LPFGM(h:1100,RTPS=0.60,t=1.0,$N_0$=5)
LPFGM(h:1100,RTPS=0.60,t=1.0,$N_0$=2)

(b) [FG] Effective Particle Size :: Temperature [K] at 500 hPa

**Figure 11:** Similar to Figure 8, but showing the LPFGM experiments with the RAOB and 1100-km localization scale. Magenta, red, blue, and green lines are LPFGM experiments with $N_0$ = 40, 10, 5, and 2, respectively. Forgetting factor $\tau$ is 1.0. The RTPS parameter is set to 1.00 in the experiment with $N_0$ = 40 (i.e., all-time resampling) and to 0.60 for in other experiments.

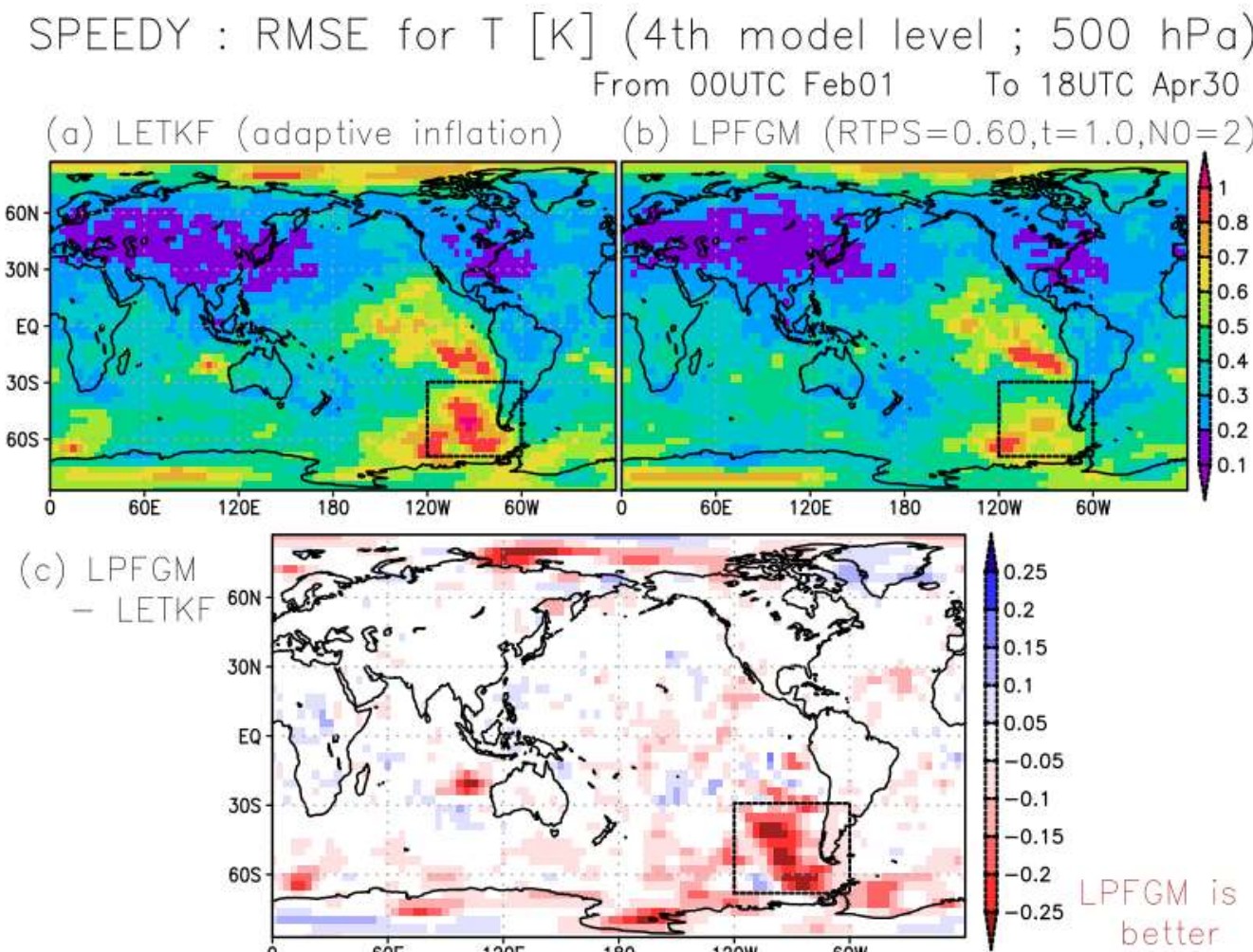

**Figure 12:** Spatial patterns of background RMSE for T (K) at the fourth model level (~500 hPa) for (a) LETKF and (b) LPFGM, averaged over February-April with RAOB and 1100-km localization scale. Panel (c) shows the difference between LETKF and LPFGM. Warm (cold) color indicates that the LPFGM is better (worse) than the LETKF. Black dashed rectangles show the region (120°W-60°W and 70°S-30°S) where difference between LETKF and LPFGM are investigated in Figure 14.

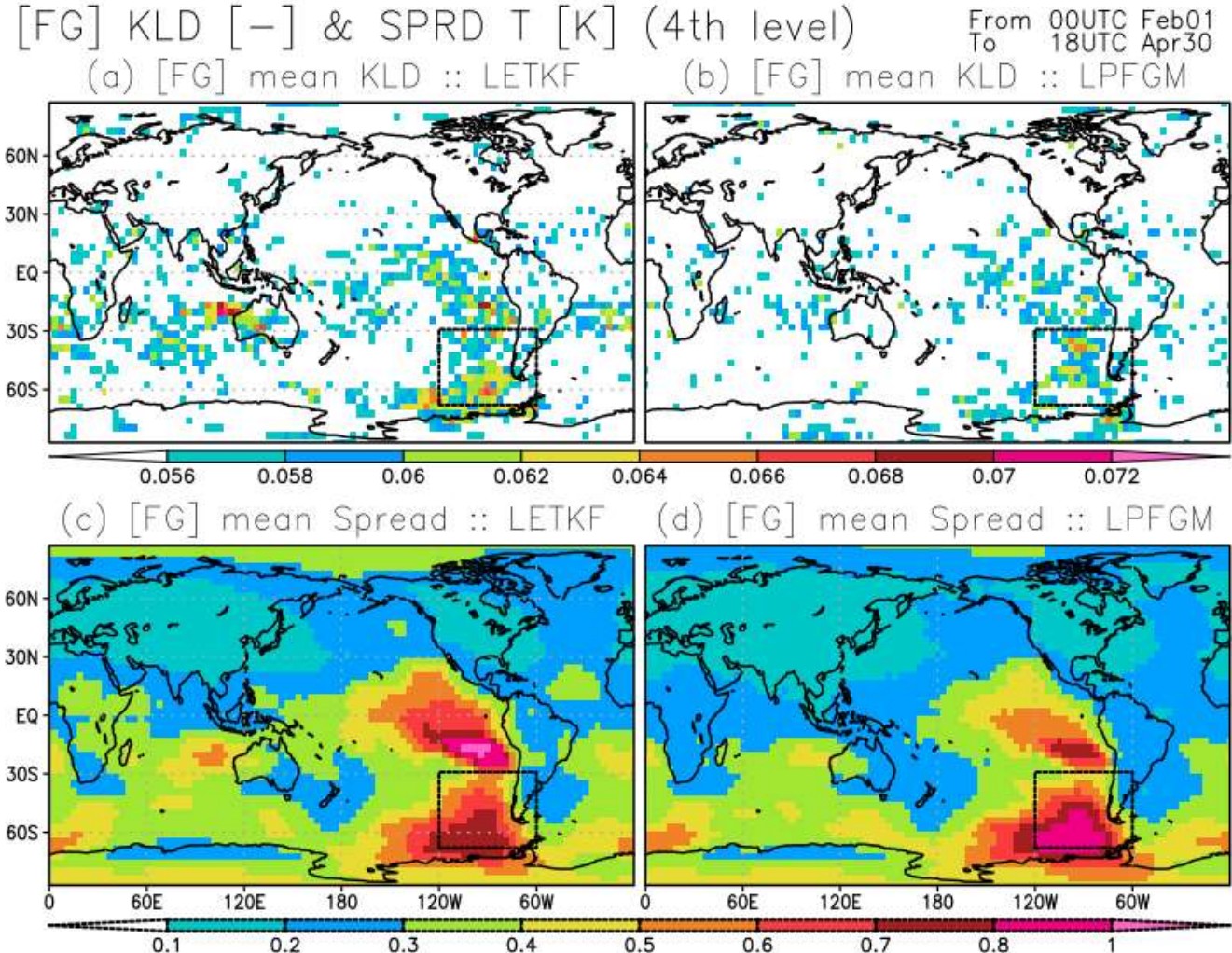

**Figure 13:** Spatial distributions of the background (a, b) Kullback–Leibler divergence and (c, d) ensemble spread for T (K), at the fourth model level (~500 hPa) averaged over February-April with RAOB and 1100-km localization scale. (a, c) and (b, d) show the LETKF and LPFGM, respectively. Black dashed rectangles show the region (120°W-60°W and 70°S-30°S) where difference between LETKF and LPFGM are investigated in Figure 14.

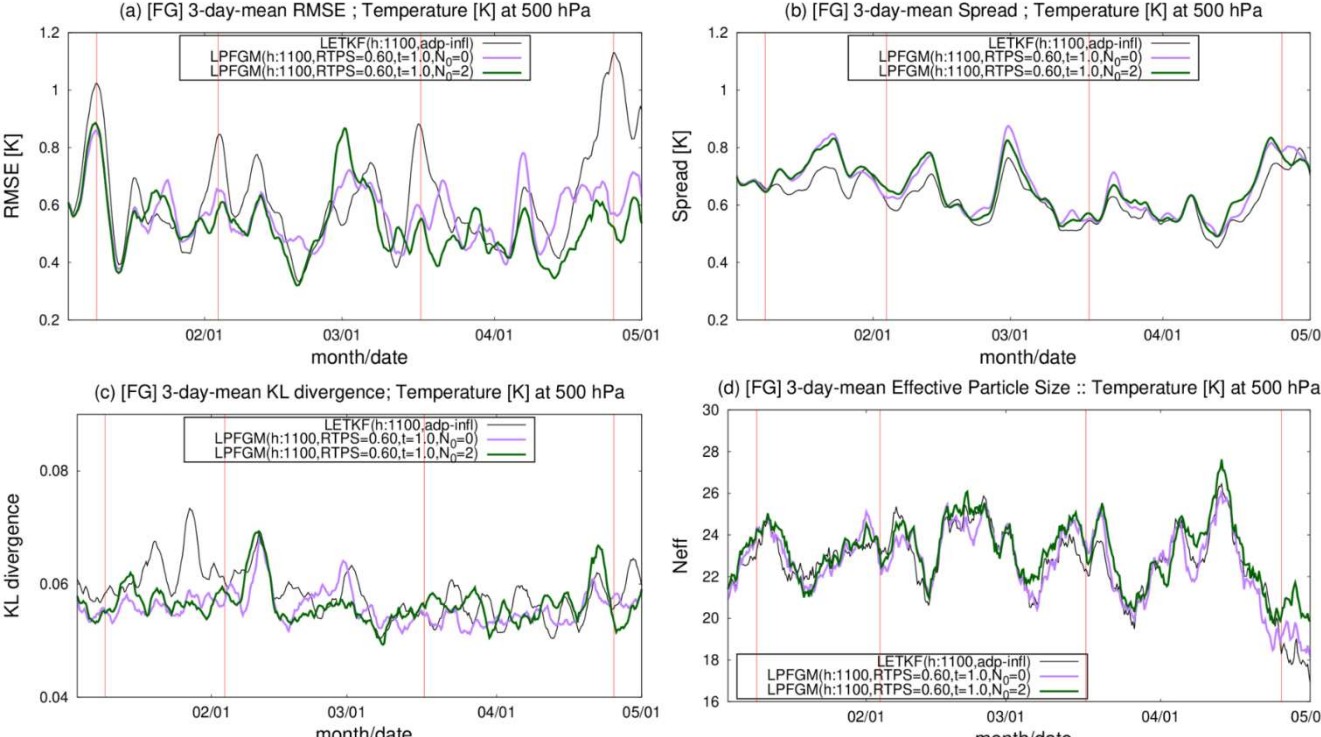

**Figure 14:** Time series of 3-day-mean (a) RMSE (K), (b) ensemble spread (K), (c) Kullback–Leibler divergence, and (d) effective particle size $N_{eff}$ at fourth model level temperature T, averaged over the region indicated by rectangles in Figs. 12 and 13 (120°W-60°W and 70°S-30°S). Black line represents the LETKF. Green and purple lines are the LPFGM whose resampling frequencies $N_0$ are 2 and 0, respectively. Vertical red lines represent the cases when the LETKF show large RMSE greater than 0.8 K. In (d), the effective particle size is also computed for the LETKF using the first-guess ensemble.

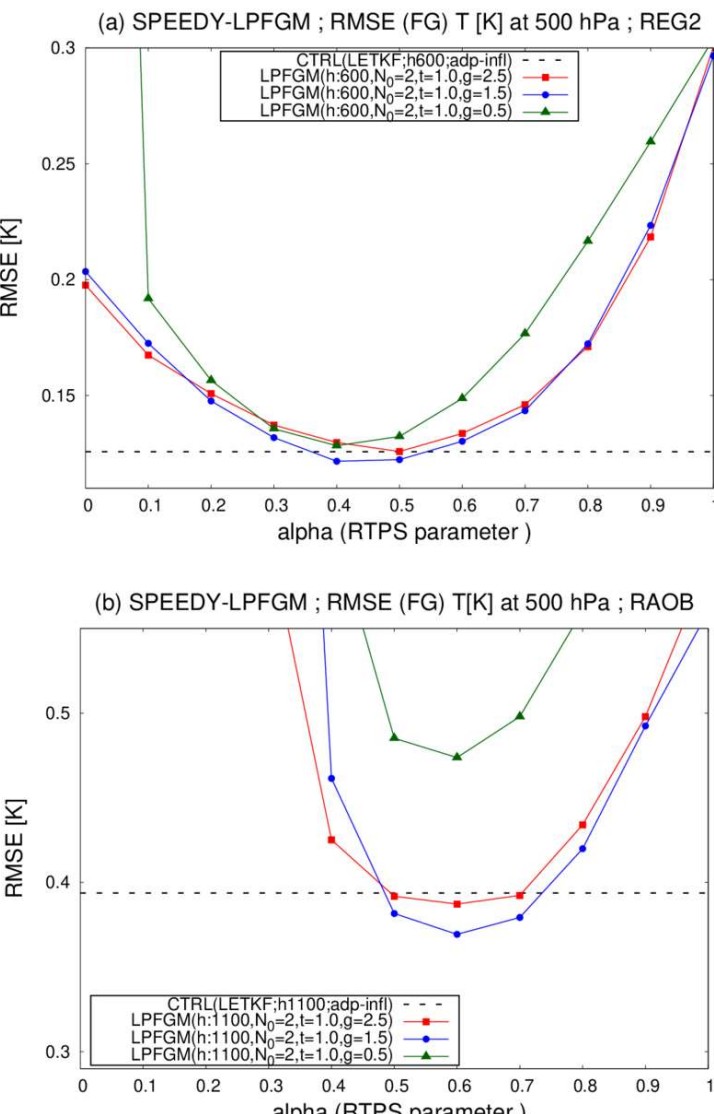

**Figure 15:** Time-mean background RMSEs for T (K) at the fourth model level (∼500 hPa) as a function of RTPS parameter $\alpha$ averaged over three months of the third year (February-April). (a) shows REG2 with 600-km localization scale, and (b) shows RAOB with 1100-km localization scale. Red, blue, and green lines are LPFGM experiments with being γ=2.5, 1.5, and 0.5, respectively. Black dashed lines represent RMSE of LETKF (0.1257 K) with adaptive multiplicative inflation instead of RTPS. Other parameters are set to be the best performing parameters: $N_0 = 2$ and $\tau = 1.0$.

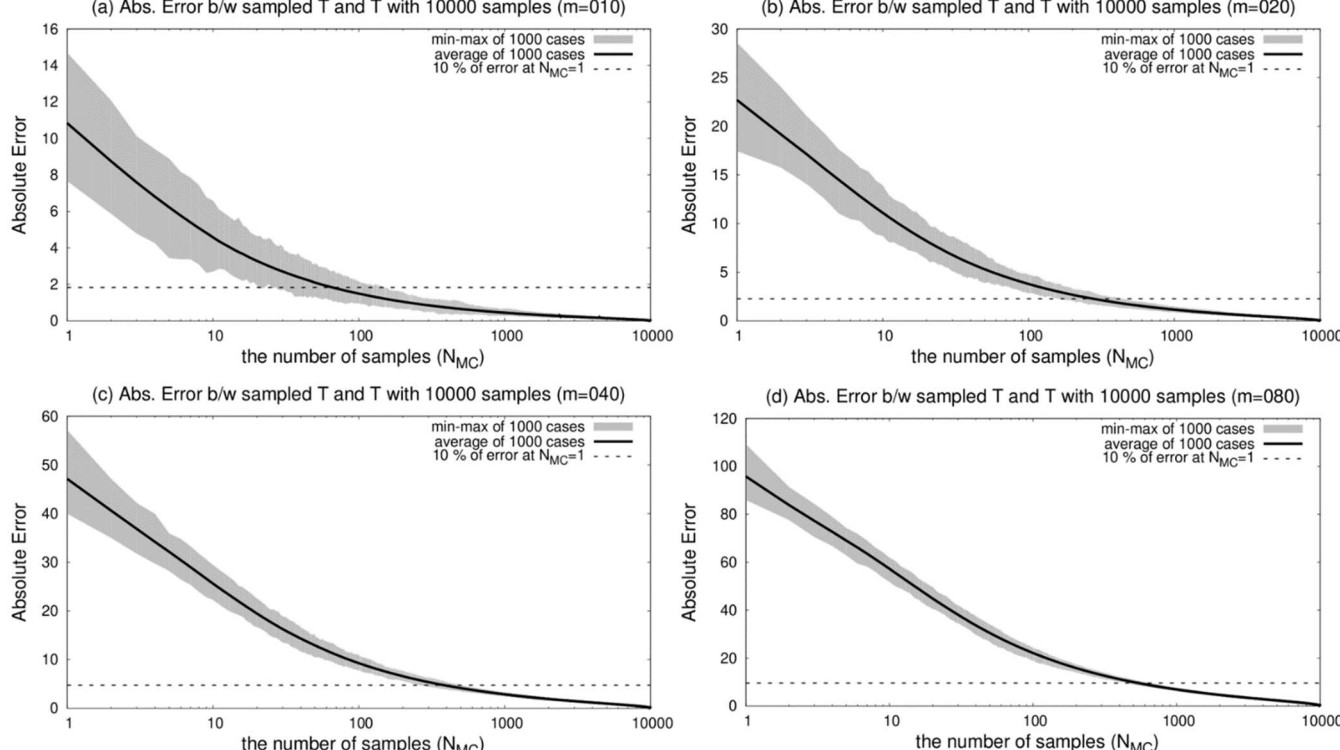

**Figure B1:** Absolute errors of the transform matrix of MC resampling as a function of the number of samples ($N_{MC}$) verified against the transform matrix with 10,000 samples. Absolute errors are computed for 1,000 independent cases that have different weights generated by uniform random numbers. Black bold lines show the average of 1,000 cases, and gray shades represent minimum and maximum errors of the 1,000 cases. Panels (a-d) show experiments with 10, 20, 40, and 80 ensembles, respectively. Dashed line shows the 10 % of initial error at $N_{MC} = 1$.
