# Peer review of "A Local Particle Filter and Its Gaussian Mixture Extension Implemented with Minor Modifications to the LETKF"

_Geoscientific Model Development, 2022_

## Author Comment (AC1)

**[Response to Reviewer's Comments]**

We are very grateful to the reviewer for her/his careful reviews and kindly giving us valuable and constructive comments and suggestions that we have generally accepted. Here, we provide our point-by-point responses to each of the comments. The revisions are highlighted by red in the revised manuscript. Supplemental PDF file would be useful to check revisions and their corresponding comments.

=======================================================================

**[Reviewer RC1]**

(1) I think this is an interesting article, which shows the potential of particle filters (and particularly two flavours of localised particle filters) in large scale atmospheric applications. The authors build on an existing LETKF computational framework to implement and experiment with these localised particle filters. The implementation is very nicely detailed with schematics and direct comparisons of the algorithms, including step-by-step contrasts, and quantifying the computational costs. I have some minor corrections and questions I would like to have answered before I can recommend this article for publication.

Response: Thank you very much for your encouraging comment. We have revised the manuscript accordingly following the suggestions.

(2) Short Summary: I would eliminate 'millions'.

Response: Revised (P1L33)

(3) Cite van Leeuwen 2021 (a consistent interpretation of the stochastic EnKF) when discussing perturbed observations.

Response: Revised (P2L39)

(4) Cite Wang et al 2004 next to Bishop 2001. After all, it is the symmetric transform variant which is used in practice.

Response: Revised (P2L50)

(5) Line 62. Even PFs with proposal densities collapse, although at a slower rate.

Response: Revised (P2L64)

(6) Line 69. How do local PFs ensure continuity between particles in different regions? Maybe mention something brief on this regard.

Response: We added discussion on this point (P5L122, P16L489)

(7) Lines 96-97. The sentence 'Transform matrix...' could be rephrased to avoid word repetition.

Response: Revised (P4L98).

(8) Line 147. Why write H(x)-y in the Gaussian likelihood. You had used y-H(x) when describing the EnKF. It is better to keep consistency in the paper.

Response: Revised as suggested (Eqs. 12 and 15)

(9) Line 205. Can there be a similar 'adaptive inflation' as Miyoshi's (and previously Anderson's) for the weights of the LPF?

Response: We added discussion on this point in section 4.3.1 (P16L472)

(10) Line 225. Does the LPFGM reduce to the LPF when ¥gamma->0?

Response: Revised (P9L235).

(11) Figure 2 is really nice an illustrative of the differences between the LPF and the LPFGM.

Response: Thank you for your comment.

(12) Line 298. There is an incomplete sentence starting with 'Although... '

Response: We are sorry the original manuscript contained this typo. This sentence was deleted in the revised manuscript (P11L318).

(13) Line 305. Are these identical twin or fraternal twin experiments? I.e., is there model error?

Response: They are identical twin experiments. We added a description (P12L324)

(14) Line 325. Why was this specific variable chosen to tune the localisation scales?

Response: We added descriptions (P11L341).

(15) Line 330. Why was T chosen? I would think that, if you want to exploit the PF advantage, you would choose a variable like relative humidity.

Response: We added explanations on this point (P13L351).

(16) Line 380. In the regular PF, the number of particles to avoid filter collapse has to be of the order exp(Neff^2), where Neff is the effective size of the problem, according to Snyder et al 2008. Then van Leeuwen and Ades (2013) showed that Neff is proportional to the number of independent observations. I therefore thought that filter collapse would happen with the sparse observation network. It is clear that filter collapse and filter divergence are different then. Could you explain the difference and the mechanisms leading to them?

Response: We added discussion on this point (P14L404).

(17) Line 390. Why does frequent resampling lead to filter divergence? Does this have to do with the space-wise continuity of the fields not being ensured?

Response: Thank you very much for the valuable comment. While we conducted further investigations, we could not find a clear reason for the filter divergence. We added related discussions in the revised manuscript (P15L422 and section 4.3.2).

---

## Author Comment (AC2)

**[Response to Reviewer's Comments]**

We are very grateful to the reviewer for her/his careful reviews and kindly giving us valuable and constructive comments and suggestions that we have generally accepted. Here, we provide our point-by-point responses to each of the comments. The revisions are highlighted by red in the revised manuscript. Supplemental PDF file would be useful to check revisions and their corresponding comments.

========================================================================

**[Reviewer RC2 General Comments]**

(1) The authors present a useful incremental step in developing a practical particle filter-based methodology that can be scaled to dimensions required for realistic operational forecasting. The results show a narrow range of parameters for the augmented LPF (i.e. the LPFGM) that can outperform a tuned LETKF implementation using a low-resolution global primitive equations model.

Response: Thank you very much for your encouraging comment. We have revised the manuscript accordingly following the suggestions.

(2) I think the manuscript would be improved if the authors could highlight a specific example case/scenario in which the dynamics require a non-Gaussian method. At its simplest, this could simple be to focus in on a regional assessment of the Southern Hemisphere East Pacific Ocean RAOB case (or the Arctic, using stenographic project map), where large differences were found between LETKF and LPFGM. It would be useful to see RMSE and ensemble spread statistics calculated for this region alone, and perhaps a bit more detailed assessment of this region as a 'case study'. This could provide further motivation for finding cases in which the LPFGM enhancement to the LETKF software architecture could provide significant value.

Response: Thank you very much for the valuable suggestion. We added results of additional investigations (P15L441; Figure 14).

(3) More strength could be given to the argument for using this method if it could be made clear that such scenarios are common occurrences in operational systems (or perhaps in reanalysis systems that focus on less observed periods in history). Are there any scenarios for a coupled Earth system forecasting system in which particular regions or physical quantities are particularly poorly observed that might benefit from this method? Do the authors see applications in reanalysis such as CERA-20C and the NOAA 20th Century Reanalysis that might improve historical reconstructions during relatively sparsely observed periods? Are there any applications in the modern satellite era where this approach could still be advantageous?

Response: thanks. We added discussion on this point (P15L455).

**[Reviewer RC2 Specific Comments]**

(1) L 19:"Therefore, implementing [the LPF] consistently with an existing LETKF code is useful."

Response: Revised (P1L19)

(2) L 21: "This study develop[s]"

Response: Revised (P1L21)

(3) L25 Change to: "The LPFGM showed more accurate and stable performance than the LPF with"

Response: Revised (P1L25)

(4) L 29: "The SPEEDY-based LETKF, LPF, and LPFGM systems [are] available"

Response: Revised (P1L29)

(5) L 37: "Ensemble[-based] data assimilation (DA)"

Response: Revised (P2L37)

(6) L37: Change to: "such as weather and ocean prediction."

Response: Revised (P2L38)

(7) L44: Change to: "models show local low dimensionality"

Response: Revised (P2L44)

(8) L44 Change to: "and practical EnKF implementations use"

Response: Revised (P2L44)

(9) L45 Change to: "that limits the impact of distant observations while also reducing the effective degrees of"

Response: Revised (P2L45)

(10) L52: issues in [the basic assumptions of the] EnKF [by permitting] nonlinear observation operator[s] and"

Response: Revised (P2L53)

(11) L57: "particles or the ensemble size [must] be increased"

Response: Revised (P2L58)

(12) L60: "equivalent-weights particle filter (ETPF"

Is this the correct acronym? It seems that EWPF would be more appropriate.

Response: Revised (P2L61)

(13) L 64: "method [for the] EnKF"

Response: Revised (P3L66)

(14) L 76-77: ", and the source code is accessible at https://github.com/takemasa-miyoshi/letkf." It seems to me that this belongs in the data availability section, or something similar, toward the end of the document.

Response: Revised as suggested (P3L68)

(15) L100: It looks a bit odd to 'divide' the matrix by beta - I think it would be more clear to write $((m-1)/beta)$ before the matrix "I".

Response: Revised (Eq. 3)

(16) L 103, equation (5): The second equality of equation (5) is perhaps not obvious. It would be helpful to add a step or two showing this relationship. (For example, showing how the relationship can be derived from the first equality in eqn. 5 and the definition in equation 6.). It seems a bit odd the way it is written here because Pa is defined using K.

Response: We added Appendix A for deriving the equations (P4P107, Appendix A).

(17) L123: "and relaxation to prior": Do you use "relaxation to prior spread", or "relaxation to prior perturbations"?

I would suggest: "and a relaxation to prior scheme (Zhang et al., 2004) as implemented by Whitaker and Hamill (2012)"

Response: Revised as suggested (P5L128)

(18) L145: It might be worth mentioning that this implies that the observation error distribution is assumed Gaussian.

Response: Revised (P6L153)

(19) L 146, equation 12: It looks like the "T" transpose operator is bolded.

Response: the transpose operator "T" was replaced to be italic and non-bolded $T$ over the entire manuscript.

(20) L167, equation 19: Could you provide some explanation of condition #1. The "m" isn't defined, nor are the indices i,j explained. The meaning of the superscripts is not explained.

Response: Revised (P6L176)

(21) L 185: "In addition, this stochastic approach approximates Eq. (19) using the Monte-Carlo approach."
Nice approach.

Response: Thank you for your comment!

(22) L185-186: "The generated transform matrix with 200 samples (Fig. 1 c) is close to that with 10,000 samples (Fig. 1 d) in the case of 40 particles." I'm curious how this Monte Carlo approach scales with the ensemble size. How does the performance (e.g. accuracy/stability of the method and the computational costs) change as the ensemble size gets large?

Response: We investigated this point in Appendix B (P7L195, Appendix B).

(23) L189: "The effective particle size N_eff"
This terminology is a bit confusing, as a particle is analogous to an ensemble member. I would either say "the effective ensemble size" or "the effective number of particles".

Response: We added descriptions (P7L197).

(24) L195, equation 23: Is this effectively a relaxation back to equal weights? I think it would be helpful to explain this more, especially since later it is indicated that only values of tau as 0 and 1 are used. What then is the implication of these two choices?

Response: Revised (P8L205)

(25) L 203: "Therefore, the LPF usually applies inflation to the posterior particles (e.g., Farchi and Bocquet 2018)." Change to: "Therefore, the LPF usually applies inflation to the posterior particles (e.g., Penny and Miyoshi, 2016; Farchi and Bocquet 2018)."

Response: Revised (P8L213).

(26) Penny and Miyoshi (2016) also applied a form of additive inflation to the posterior: "at each cycle we add Gaussian noise with variance scaled locally to a magnitude matching the analysis error variance and apply this to each analysis member prior to the subsequent ensemble forecast. The amplitude of the additive noise was chosen to conform to the dynamics of the growing error subspace, as estimated by the analysis ensemble spread." (Penny and Miyoshi, 2016)

Response: We added descriptions on this point in section 4.3.1 (P16L476)

(27) L 206: "based on [the] authors' preliminary experiments"

Response: Revised (P8L216)

(28) L214: "The Gaussian mixture extension of the LPF is [one] such hybrid algorithm"

Response: Revised (P8L225)

(29) L218: "hut" Change to "hat"

Response: Revised (P8L229)

(30) L223: How does increasing gamma reduce the amplitude? Is there a relationship between gamma and the number of ensemble members?

Response: We added descriptions and discussion on this point (P8L234, P17L502).

(31) L 226, equation 26: Am I interpreting correctly that the Kalman filter is applied to each particle independently, using the forecast error covariance estimated from the entire ensemble? Or, is it applied to the adjust the mean of the individual particles like an ETKF?

Response: Yes, you are right. We added explanations (P9L243).

(32) L228: I see that $K^\wedge$ is defined here, but I do not see a definition for $Pa^\wedge$. It seems that $Pb^\wedge$ is a scaled version of the ensemble forecast error covariance, but I'd like to see a more precise discussion of how $K^\wedge$ is formed - is it the same for every particle, or does each particle have a different $K^\wedge$? If all of the particles have the same $K^\wedge$, then how is this different than applying the standard EnKF update to the mean and perturbations? Please provide more discussion on these points.

Response: We added descriptions on this point (P9L243).

(33) L235: Do you mean $T_{t,GM}$?

Response: Revised (P9L250).

(34) L 241: What is WP22? Can you just provide the full reference?

Response: WP22 (Walter and Potthast 2022) is defined in earlier part of the manuscript (P3L73).

(35) L 247, equation 33: In practice, would there be any benefit to forming and applying the $T_{t,LPFGM}$ matrix directly?

Response: We added explanations on this point (P10L263).

(36) L249: "may appear [to use] the same observations twice"

Response: Revised (P10L268).

(37) L290: You might reiterate here that typically $N\_MC << p\_L$.

Response: Revised (P11L309).

(38) L298: "Although running the SPEEDY model is, the model contains The SPEEDY model"
This looks like some kind of typo or grammatical mistake.

Response: We are sorry the original manuscript contained this typo. This sentence was deleted in the revised manuscript (P11L318).

(39) L 307: "Gaussian noises were added" Change to: "Gaussian noise was added"
Response: Revised (P12L327).

(40) L308: "data whose interval is 6 h" Change to: "data at 6 h intervals"
Response: Revised (P11L328).

(41) L311: "every seven layers" Change to: "all seven layers"
Response: Revised (P11L331).

(42) L311-312: "The ensemble size is 40 and their initial conditions were taken from an independent nature run." This is not clear - was a free-running ensemble used as the nature run? Or was the initial ensemble sampled from a single deterministic nature run? Please be more precise.
Response: Revised (P11L332).

(43) L320: "LETKF were optimized manually" Change to: "LETKF were tuned manually"
If this was done manually, then it is likely not optimized.
Response: Revised (P11L340).

(44) L417-418: "An alternative method of inflation is adding random noise to the transform matrix (Potthast et al., 2019). However, regulating the amplitude of the random noise was not trivial in the authors' preliminary experiments with L96 (not shown)." Have the authors tried inflation method as detailed by Penny and Miyoshi (2016). There the amplitude was specified by the local analysis error variance, and could be interpreted as noise applied to the transform matrix (it is applied to the analysis perturbations).
Response: We have not tried the approach, and thank you for your valuable suggestion. We added discussion on this point (P16L476).

(45) L446: "This study considered two experimental settings with and without weight succession" It seems like at least one case in between (e.g. tau = 0.5) should be tried to give some idea of the benefit of this weighting method.
Response: We added experiments, and discussion with different tau (P13L377; Figure 7).

(46) L 456: "optimal localization scale " Again, if not mathematically optimized, then I would change this to: "best performing localization scale"
Response: Revised for entire manuscript.

(47) L461: "forecast in a sparsely observed regions" Change to: "forecast in sparsely observed regions"

Response: Revised (P18L518)

(48) L700, figure 9: Please use white instead of yellow for the "near zero" values in panels (d) and (e). (Same comment for Figure 12)

Response: Revised as suggested (Figures 9 and 12).

---

## Author Response (AR2)

**[Comments to Editor]**

We are very grateful to the reviewer for her/his careful reviews and kindly giving us valuable and constructive comments. Here, we provide our point-by-point minor revisions for final publications.

================================================================================

(1) Equation 14 was slightly modified for improving readability of equations.

(2) from "previous PDF" to "prior PDF" (P8L225)

(3) Equation (28) and P9L240 : from $\widetilde{\mathbf{P}}_t^a$ to $\widehat{\mathbf{P}}_t^a$

(4) P10L266: We added "for LPFs"

(5) P16L466: from "Kalman filter" to "Kalman gain"

(6) P19L562: We added "Competing interests"

(7) P25: Algorithm 1 was slightly updated for improving readability

(8) P26L734: We added "for generating"

(9) Caption of Figure 2: We modified a typo.

(10) Caption of Figure 11: We added additional descriptions to specify experimental settings.